# $\mathcal{X}$-Scene: Large-Scale Driving Scene Generation with High Fidelity and Flexible Controllability

Yu Yang[1,2]    Alan Liang[2]    Jianbiao Mei[1]    Yukai Ma[1]    Yong Liu[1,†]    Gim Hee Lee[2,†]

[1] Zhejiang University    [2] National University of Singapore

https://x-scene.github.io/

## Abstract

Diffusion models are advancing autonomous driving by enabling realistic data synthesis, predictive end-to-end planning, and closed-loop simulation, with a primary focus on temporally consistent generation. However, large-scale 3D scene generation requiring spatial coherence remains underexplored. In this paper, we present $\mathcal{X}$-Scene, a novel framework for large-scale driving scene generation that achieves geometric intricacy, appearance fidelity, and flexible controllability. Specifically, $\mathcal{X}$-Scene supports multi-granular control, including low-level layout conditioning driven by user input or text for detailed scene composition, and high-level semantic guidance informed by user intent and LLM-enriched prompts for efficient customization. To enhance geometric and visual fidelity, we introduce a unified pipeline that sequentially generates 3D semantic occupancy and corresponding multi-view images and videos, ensuring alignment and temporal consistency across modalities. We further extend local regions into large-scale scenes via consistency-aware outpainting, which extrapolates occupancy and images from previously generated areas to maintain spatial and visual coherence. The resulting scenes are lifted into high-quality 3DGS representations, supporting diverse applications such as simulation and scene exploration. Extensive experiments demonstrate that $\mathcal{X}$-Scene substantially advances controllability and fidelity in large-scale scene generation, empowering data generation and simulation for autonomous driving.

## 1 Introduction

Recent advancements in generative AI have profoundly impacted autonomous driving, with diffusion models (DMs) emerging as pivotal tools for data synthesis and driving simulation. Some approaches utilize DMs as data machines, producing high-fidelity driving videos [1–14] or multi-modal synthetic data [15–18] to augment perception tasks, as well as generating corner cases (e.g., vehicle cut-ins) to enrich planning data with uncommon yet critical scenarios. Beyond this, other methods employ DMs as world models to predict future driving states, enabling end-to-end planning [19–21] and closed-loop simulation [22–28]. All these efforts emphasize ***long-term video generation through temporal recursion***, encouraging DMs to produce coherent video sequences for downstream tasks.

However, ***large-scale scene generation with spatial expansion***, which aims to build expansive and immersive 3D environments for arbitrary driving simulation, remains an emerging yet underexplored direction. A handful of pioneering works have explored 3D driving scene generation at scale. For example, SemCity [29] generates city-scale 3D occupancy grids using DMs, but the lack of appearance details limits its practicality for realistic simulation. UniScene [18] and InfiniCube [30] extend this by generating both 3D occupancy and images, but require a manually defined large-scale layout as a conditioning input, complicating the generation process and hindering flexibility.

---

† corresponding author

39th Conference on Neural Information Processing Systems (NeurIPS 2025).

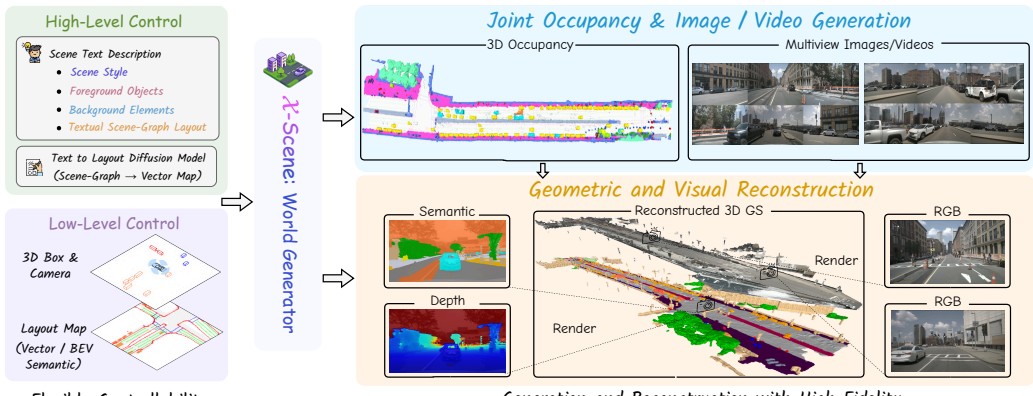

Figure 1: **Overview of 𝒳-Scene**, a unified world generator that supports *multi-granular controllability* through high-level text-to-layout generation and low-level BEV layout conditioning. It performs *joint occupancy, image, and video generation* for 3D scene synthesis and reconstruction with high fidelity.

In this work, we tackle the problem of large-scale scene generation with spatial expansion, which presents three key challenges: 1) *Flexible Controllability*: enabling versatile control through both low-level conditions (e.g., layouts) for precise scene composition and high-level prompts (e.g., user-intent text descriptions) for intuitive customization; 2) *High-Fidelity Geometry and Appearance*: generating intricate geometry with photorealistic appearance to ensure structural integrity and visual realism in 3D scenes; 3) *Large-Scale Consistency*: maintaining spatial coherence across extended regions to ensure global consistency throughout the generated environment.

To address these challenges, we propose 𝒳*-Scene*, a novel framework for large-scale driving scene generation featuring: **1) *Multi-Granular Controllability*:** It enables users to guide generation at multiple abstraction levels, supporting fine-grained BEV semantic layouts for precise control and high-level text prompts for efficient customization. Text prompts are enriched by LLMs into detailed scene narratives, structured as scene graphs and converted into vector-map layouts via a scene-graph to layout diffusion module. These layouts provide spatial and semantic cues that guide subsequent scene synthesis, combining layout-level precision with prompt-based flexibility. **2) *Geometric and Visual Fidelity*:** 𝒳*-Scene* employs a unified pipeline that sequentially generates 3D semantic occupancy and corresponding multi-view images and videos, ensuring structural accuracy, photorealistic appearance, and temporal consistency with cross-modal alignment. **3) *Consistent Large-Scale Extrapolation*:** To synthesize expansive environments, it progressively extrapolates new scene content conditioned on adjacent, previously generated regions. The consistency-aware outpainting mechanism preserves spatial continuity and enables seamless extension beyond local areas.

Furthermore, to support downstream applications such as realistic driving simulation, we reconstruct the generated occupancy and multi-view images/videos into 3D Gaussian (3DGS) [31] representations, which faithfully preserve geometric detail and visual quality. By unifying controllability, fidelity, and scalability, 𝒳*-Scene* advances the state-of-the-art in large-scale, controllable driving scene synthesis, empowering realistic data generation and simulation for autonomous driving.

The main contributions of our work are summarized as follows:

- We propose 𝒳*-Scene*, a novel framework for large-scale 3D driving scene generation with multi-granular controllability, geometric and visual fidelity, and consistent large-scale extrapolation, supporting a wide range of downstream applications.

- We design a flexible multi-granular control mechanism that synergistically combines high-level semantic guidance (LLM-enriched text prompts) with low-level geometric specifications (user-provided or text-driven layout), enabling scene creation tailored to diverse user needs.

- We present a unified occupancy–image–video generation pipeline that achieves geometric fidelity, photorealistic appearance, and temporal coherence, enabling seamless large-scale scene expansion.

- Extensive experiments show 𝒳*-Scene* achieves superior performance in generation quality and controllability, enabling diverse applications from data augmentation to driving simulation.

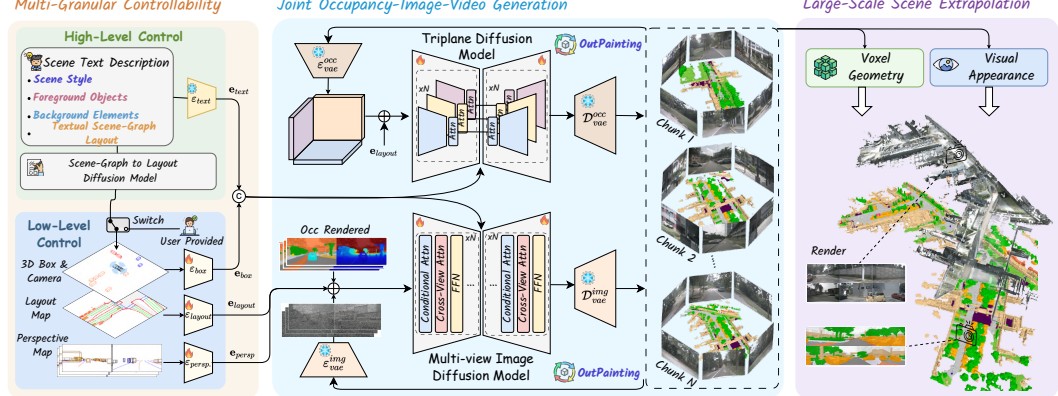

Figure 2: **Pipeline of 𝒳-Scene for driving scene generation:** (a) *Multi-granular controllability* supports both high-level text prompts and low-level geometric constraints for flexible specification; (b) *Joint occupancy-image-video generation* synthesizes aligned 3D voxels and multi-view images and videos via conditional diffusion; (c) *Large-scale extrapolation* enables coherent scene expansion through consistency-aware outpainting (Fig. 4). Fig. 3 details the scene-graph to layout diffusion.

## 2 Related Works

**Driving Image and Video Generation.** Diffusion models [32–35] have revolutionized image synthesis by iteratively refining Gaussian noise into high-quality results. Building on this, they have greatly advanced autonomous driving by enabling realistic image and video generation for various downstream tasks. Several methods synthesize driving images [1, 36–39] or videos [2–14] from layout conditions to augment perception data. Others [40, 41] generate rare yet critical events, e.g., lane changes or cut-ins, to improve planning under corner cases. Moreover, diffusion-based world models predict future driving videos for end-to-end planning [19–21] or closed-loop simulation [22–27]. While prior works emphasize temporal consistency, our approach explores the complementary aspect of spatial coherence for large-scale scene generation.

**3D and 4D Driving Scene Generation.** Recent advances extend beyond 2D generation to 3D/4D scene synthesis [42], producing 3D environments from LiDAR point clouds [43–52], occupancy volumes [53, 54, 29, 55–59], or 3D Gaussian Splatting (3DGS) [60–67], serving as neural simulators for data generation and driving simulation. The field has futher progressed in two directions: 1) 3D world models that predict future scene representations (e.g., point clouds [68–70] or occupancy maps [71–76]) to aid planning and pretraining; and 2) multi-modal generators that synthesize aligned data across modalities, such as image–LiDAR [15, 16] or image–occupancy pairs [17, 18, 24]. Our work explores joint occupancy–image–video generation, constructing scenes that integrate fine-grained geometry, photorealistic appearance, and temporally coherent dynamics.

**Large-Scale Scene Generation.** Large-scale city generation has evolved along four main directions: video-based [77, 78], outpainting-based [79–81], PCG-based [82–84], and neural-based methods [85–87]. While effective for natural or urban environments, these approaches are not tailored for driving scenarios requiring accurate street layouts and dynamic agents. Driving-specific methods also face key limitations: XCube [58] and SemCity [29] model only geometric occupancy without appearance, while DrivingSphere [24], UniScene [18], and InfiniCube [30] depend on manually designed large-scale layouts, limiting scalability. In contrast, our 𝒳-Scene framework jointly generates geometry and appearance with flexible, text-driven control, offering efficient and user-friendly customization.

## 3 Methodology

𝒳-*Scene* aims to generate large-scale 3D driving scenes within a unified framework addressing controllability, fidelity, and scalability. As shown in Fig. 2, it consists of three main components: **1) Multi-Granular Controllability** (Sec. 3.1), which integrates high-level user intent with low-level geometric constraints for flexible scene specification; **2) Joint Occupancy, Image, and Video Generation** (Sec. 3.2), which employs conditioned diffusion models to synthesize 3D voxel occupancy, multi-view images, and temporally coherent videos with 3D-aware guidance; and **3) Large-Scale**

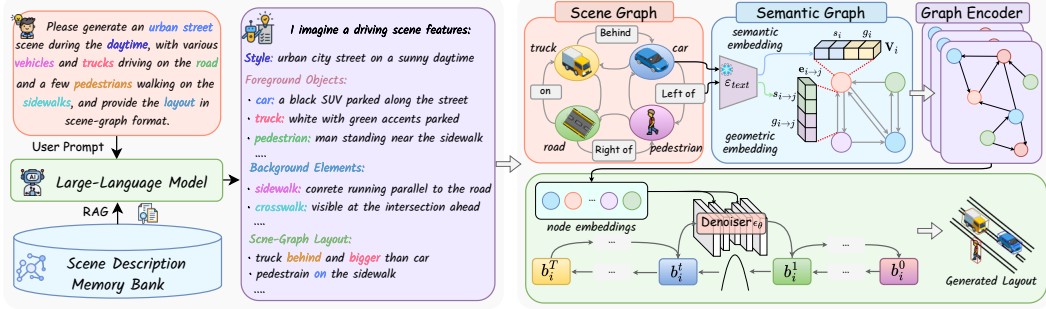

(a) Textual Description Enrichment          (b) Textual Scene-Graph to Layout Generation

Figure 3: **Pipeline of textual description enrichment and scene-graph to layout generation**: (a) Input prompts are enriched using RAG-augmented LLMs to produce structured scene descriptions; (b) Spatial relationships are converted into a scene graph and encoded with a graph network, followed by conditional diffusion that denoises object boxes and lane polylines into the final layouts.

**Scene Extrapolation and Reconstruction** (Sec. 3.3), which extends scenes via consistency-aware outpainting and lifts them into 3DGS representations for downstream simulation and exploration.

## 3.1 Multi-Granular Controllability

$\mathcal{X}$-*Scene* supports dual-mode scene control through: 1) high-level textual prompts, which are enriched by LLMs and converted into structured layouts via a text-to-layout generation model (illustrated in Fig. 3); and 2) direct low-level geometric control for precise spatial specification. This hybrid approach enables both intuitive creative expression and exacting scene customization.

**Text Description Enrichment.** Given a coarse user-provided textual prompt $\mathcal{T_P}$, we first enrich it into a comprehensive scene description $\mathcal{D} = \{\mathcal{S}, \mathcal{O}, \mathcal{B}, \mathcal{L}\}$, comprising: scene style $\mathcal{S}$ (weather, lighting, environment), foreground objects $\mathcal{O}$ (semantics, spatial attributes, and appearance), background elements $\mathcal{B}$ (semantics and visual characteristics), and textual scene-graph layout $\mathcal{L}$, representing spatial relationships among scene entities. The structured description $\mathcal{D}$ is generated as:

$$\mathcal{D} = \mathcal{G}_{\text{description}}\big(\mathcal{T_P}, \text{RAG}(\mathcal{T_P}, \mathcal{M})\big) \tag{1}$$

where $\mathcal{M} = \{m_i\}_{i=1}^N$ denotes the scene description memory. Each entity $m_i$ is automatically constructed using one of the collected scene datasets by: 1) extracting $\{\mathcal{S}, \mathcal{O}, \mathcal{B}\}$ using VLMs on scene images; and 2) converting spatial annotations (object boxes and road lanes) into textual scene-graph layout $\mathcal{L}$. As shown in Fig. 3, the Retrieval-Augmented Generation (RAG) module retrieves relevant descriptions similar to $\mathcal{T_P}$ from the memory bank $\mathcal{M}$, which are then composed into a detailed, user-intended scene description by an LLM-based generator $\mathcal{G}_{\text{description}}$.

This pipeline leverages RAG for few-shot retrieval and composition when processing brief user prompts, enabling flexible and context-aware scene synthesis. The memory bank $\mathcal{M}$ is designed to be extensible, allowing seamless integration of new datasets to support a broader variety of scene styles. Additional examples of generated scene descriptions are provided in the appendix.

**Textual Scene-Graph to Layout Generation.** Given the textual layout $\mathcal{L}$, we follow prior works [88, 89] and translate it into a spatial layout map via a scene-graph–to–layout pipeline (Fig. 3). First, we construct a scene graph $\mathcal{G} = (\mathcal{V}, \mathcal{E})$, where nodes $\mathcal{V} = \{v_i\}_{i=1}^M$ represent $M$ scene entities (e.g., *cars*, *pedestrians*, *road lanes*) and edges $\mathcal{E} = \{e_{i \to j} | i, j \in \{1, ..., M\}\}$ represent spatial relations (e.g., *front of*, *on top of*). Each node and edge is then embedded by concatenating semantic features $s_i$, $s_{i \to j}$ (extracted via a text encoder $\mathcal{E}_{\text{text}}$) with learnable geometric features $g_i$, $g_{i \to j}$, resulting in node embeddings $\mathbf{v}_i = \text{Concat}(s_i, g_i)$ and edge embeddings $\mathbf{e}_{i \to j} = \text{Concat}(s_{i \to j}, g_{i \to j})$.

The graph embeddings are refined using a graph convolutional network, which propagates contextual information $\mathbf{e}_{i \to j}$ across the graph and updates each node embedding $\mathbf{v}_i$ via neighborhood aggregation. Finally, layout generation is formulated as a conditional diffusion process: each object layout is initialized as a noisy 7-D vector $b_i \in \mathbb{R}^7$ (representing box center, dimensions, and orientation), while each road lane begins as a set of $N$ noisy 2D points $p_i \in \mathbb{R}^{N \times 2}$, with denoising process is conditioned on the corresponding node embeddings $\mathbf{v}_i$ to produce geometrically coherent placements.

**Low-Level Conditional Encoding.** We encode fine-grained conditions (such as user-provided or model-generated layout maps and 3D bounding boxes) into embeddings to enable precise geometric control. As illustrated in Fig. 2, the 2D layout maps are processed by a ConvNet ($\mathcal{E}_{layout}$) to extract layout embeddings $\mathbf{e}_{layout}$, while 3D box embeddings $\mathbf{e}_{box}$ are obtained via MLPs ($\mathcal{E}_{box}$), which fuse object class and spatial coordinate features. To further enhance geometric alignment, we project both the scene layout and 3D boxes into the camera view to generate perspective maps, which are encoded by another ConvNet ($\mathcal{E}_{persp.}$) to capture spatial constraints from the image plane. Additionally, high-level scene descriptions $\mathcal{D}$ are embedded via a T5 encoder ($\mathcal{E}_{text}$), providing rich semantic cues for controllable generation through the resulting text embeddings $\mathbf{e}_{text}$.

## 3.2 Joint Occupancy, Image, and Video Generation

We adopt a joint 3D-to-2D generation hierarchy that first models scene geometry via occupancy diffusion, followed by image synthesis guided by occupancy-rendered maps to ensure geometric consistency. The pipeline is further extended with a temporal diffusion module for video generation, producing smooth motion and cross-view temporal coherence.

**Occupancy Generation via Triplane Diffusion.** We adopt a triplane representation [90] to encode 3D occupancy fields with high geometric fidelity. Given an occupancy volume $\mathbf{o} \in \mathbb{R}^{X \times Y \times Z}$, a triplane encoder compresses it into three orthogonal latent planes $\mathbf{h} = \{\mathbf{h}^{xy}, \mathbf{h}^{xz}, \mathbf{h}^{yz}\}$ with spatial downsampling. To mitigate information loss due to reduced resolution, we propose a novel triplane deformable attention mechanism that aggregates richer features for a query point $\mathbf{q} = (x, y, z)$ as:

$$\mathbf{F_q}(x,y,z) = \sum_{\mathcal{P} \in \{xy, xz, yz\}} \sum_{k=1}^{K} \sigma\big(\mathbf{W}_\omega^\mathcal{P} \cdot \text{PE}(x,y,z)\big)_k \cdot \mathbf{h}^\mathcal{P}\Big(\text{proj}_\mathcal{P}(x,y,z) + \Delta p_k^\mathcal{P}\Big) \quad (2)$$

where $K$ is the number of sampling points, $\text{PE}(\cdot) : \mathbb{R}^3 \to \mathbb{R}^D$ denotes positional encoding, and $\mathbf{W}_\omega^\mathcal{P} \in \mathbb{R}^{K \times D}$ generates attention weights with the softmax function $\sigma(\cdot)$. The projection function $\text{proj}_\mathcal{P}$ maps 3D coordinates to 2D planes (e.g., $\text{proj}_{xy}(x,y,z) = (x,y)$), and the learnable offset $\Delta p_k^\mathcal{P} = \mathbf{W}_o^\mathcal{P}[k] \cdot \text{PE}(x,y,z) \in \mathbb{R}^2$ uses weights $\mathbf{W}_o^\mathcal{P} \in \mathbb{R}^{2 \times D}$ to shift sampling positions for better feature alignment. Then the triplane-VAE decoder reconstructs the 3D occupancy field from the aggregated features $\mathbf{F_q}$.

Building on the latent triplane representation $\mathbf{h}$, we introduce a conditional diffusion model $\epsilon_\theta^{occ}$ that synthesizes novel triplanes through iterative denoising. At each timestep $t$, the model refines a noisy triplane $\mathbf{h}_t$ toward the clean target $\mathbf{h}_0$ using two complementary conditioning strategies: 1) additive spatial conditioning with the layout embedding $\mathbf{e}_{layout}$; and 2) cross-attention-based conditioning with $\mathcal{C} = \text{Concat}(\mathbf{e}_{box}, \mathbf{e}_{text})$, integrating geometric and semantic constraints. The model is trained to predict the added noise $\epsilon$ using the denoising objective: $\mathcal{L}_{diff}^{occ} = \mathbb{E}_{t,\mathbf{h}_0,\epsilon} \left[\|\epsilon - \epsilon_\theta^{occ}(\mathbf{h}_t, t, \mathbf{e}_{layout}, \mathcal{C})\|_2^2\right]$.

**Image Generation with 3D Geometry Guidance.** After obtaining the 3D occupancy, we convert voxels into 3D Gaussian primitives parameterized by voxel coordinates, semantics, and opacity, which are rendered into semantic and depth maps via tile-based rasterization [31]. To incorporate object-level geometry, we first generate normalized 3D coordinates for the entire scene and extract object-specific regions based on bounding boxes. The corresponding coordinates are encoded into object positional embeddings $\mathbf{e}_{pos}$, providing fine-grained geometric guidance. The semantic, depth, and layout (or perspective) maps are processed by ConvNets and fused with $\mathbf{e}_{pos}$ to form the final geometric embedding $\mathbf{e}_{geo}$. This embedding is combined with noisy image latents to achieve pixel-aligned geometric conditioning. The image diffusion model $\epsilon_\theta^{img}$ further leverages cross-attention with conditions $\mathcal{C}$ (text, camera, and box embeddings) for appearance control. The model is trained via: $\mathcal{L}_{diff}^{img} = \mathbb{E}_{t,\mathbf{x}_0,\epsilon} \left[\|\epsilon - \epsilon_\theta^{img}(\mathbf{x}_t, t, \mathbf{e}_{geo}, \mathcal{C})\|_2^2\right]$.

**Video Generation with Motion-Aware Diffusion.** After obtaining multi-view images, we extend the diffusion framework to synthesize temporally coherent videos conditioned on motion cues. The generated images from preceding clips serve as *reference frames* to guide the denoising of subsequent noisy latents $\mathbf{x}_t$. The diffusion model $\epsilon_\theta^{vid}$ takes both $\mathbf{x}_t$ and encoded reference features $\mathbf{F}_{ref}$, concatenated along the temporal dimension, and applies a temporal self-attention layer to capture motion correspondences, with the relative ego poses $\mathbf{P}_{rel}$ also encoded for motion-aware conditioning.

Only the temporal attention layers are fine-tuned from the pre-trained image diffusion model, enabling efficient transfer from spatial to temporal domains. The training objective follows the denoising formulation: $\mathcal{L}_{\text{diff}}^{\text{vid}} = \mathbb{E}_{t,\mathbf{x}_0,\epsilon}[\|\epsilon - \epsilon_\theta^{\text{vid}}(\mathbf{x}_t, t, \mathbf{F}_{\text{ref}}, \mathbf{P}_{\text{rel}}, \mathcal{C})\|_2^2]$. During inference, an *autoregressive* strategy is employed for streaming video generation, where previously generated frames are reused as motion references to ensure smooth transitions and temporal coherence across clips.

### 3.3 Large-Scale Scene Extrapolation and Reconstruction

Building on single-chunk generation, we propose a progressive extrapolation approach that coherently expands occupancy and images across multiple chunks, maintaining geometric and visual consistency with the generated multi-view videos for downstream applications.

**Geometry-Consistent Scene Outpainting.** We extend the occupancy field via triplane extrapolation [91], which decomposes the task into extrapolating three orthogonal 2D planes, as illustrated in Fig. 4. The core idea is to generate a new latent plane $\mathbf{h}_0^{\text{new}}$ by synchronizing its denoising process with the forward diffusion of a known reference plane $\mathbf{h}_0^{\text{ref}}$, guided by an overlap mask $\mathbf{M}$. Specifically, at each denoising step $t$, the new latent is updated as:

$$\mathbf{h}_{t-1}^{\text{new}} \leftarrow \left(\sqrt{\bar{\alpha}_t}\mathbf{h}_0^{\text{ref}} + \sqrt{1 - \bar{\alpha}_t}\boldsymbol{\epsilon}\right) \odot \mathbf{M} + \epsilon_\theta^{occ}(\mathbf{h}_t^{\text{new}}, t) \odot (1 - \mathbf{M}) \tag{3}$$

where $\boldsymbol{\epsilon} \sim \mathcal{N}(\mathbf{0}, \mathbf{I})$ and $\bar{\alpha}_t$ is determined by the noise scheduler at timestep $t$. This method preserves structural consistency in overlapping regions while plausibly extending reference content into unseen areas, yielding coherent and geometry-consistent scene extensions.

**Visual-Coherent Image Extrapolation.** Beyond occupancy outpainting, we further extrapolate the visual field for synchronized image generation. To maintain visual coherence between the reference image $\mathbf{x}_0^{\text{ref}}$ and the new view $\mathbf{x}_0^{\text{new}}$, a naive approach warps $\mathbf{x}_0^{\text{ref}}$ using the camera pose $(R, T)$ and applies image inpainting (Fig. 4). However, using only warped images as conditions is inadequate. To address this, we fine-tune the diffusion model $\epsilon_\theta^{\text{img}}$ with explicit conditioning on $\mathbf{x}_0^{\text{ref}}$ and camera embeddings $\mathbf{e}(R, T)$. Concretely, $\mathbf{x}_0^{\text{ref}}$ is concatenated with the novel latent $\mathbf{x}_t^{\text{new}}$, and $\mathbf{e}(R, T)$ is incorporated via cross-attention, enabling view-consistent extrapolation with photorealistic visual results.

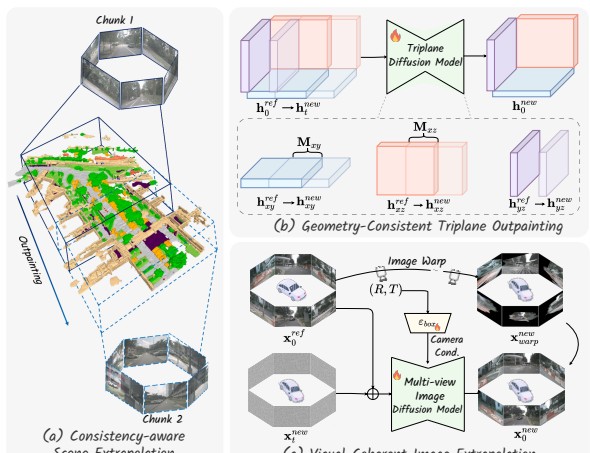

Figure 4: **Illustration of (a) consistency-aware outpainting:** (b) Occupancy triplane extrapolation is decomposed into three 2D plane extensions guided by overlapped regions; (c) Image extrapolation is performed via diffusion conditioned on images and camera parameters.

## 4 Experiments

### 4.1 Experimental Settings

We use Occ3D-nuScenes [92] to train the occupancy module and nuScenes [93] for the multi-view image and video generation modules. Additional implementation details are provided in the appendix.

**Experimental Tasks and Metrics.** We evaluate $\mathcal{X}$-*Scene* across three aspects using a range of metrics: **1) Occupancy Generation**: We evaluate the reconstruction results of the VAE with IoU and mIoU metrics. For occupancy generation, following [59], we report both generative 3D and 2D metrics, including Inception Score, FID, KID, Precision, Recall, and F-Score. **2) Multi-View Image Generation**: We evaluate the quality of the synthesized images using FID. **3) Multi-View Video Generation**: We evaluate video temporal consistency using FVD. **4) Downstream Tasks**: We evaluate the sim-to-real gap by measuring performance on the generated scenes across downstream

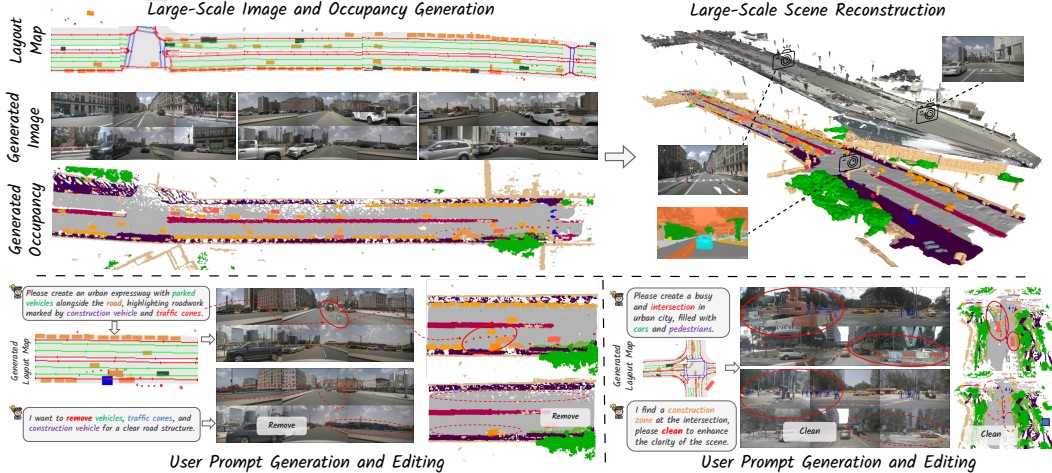

Figure 5: **Versatile generation capability of 𝒳-Scene:** (a) Generation of large-scale, consistent semantic occupancy and multi-view images, which are reconstructed into 3D scenes for multi-view rendering; (b) User-prompted layout and scene generation, along with scene geometry editing.

Table 1: **Comparisons of occupancy reconstruction of the VAE.** The downsampled size is reported in terms of spatial dimensions (H, W) and feature dimension (C).

| Method | OccSora [72] (VQVAE) | OccWorld [71] (VQVAE) | OccLLama [94] (VQVAE) | UniScene [18] (VAE) | 𝒳-Scene (Ours) (Triplane-VAE) | |
|---|---|---|---|---|---|---|
| **Downsampled Size** | (T/8,25,25,512) | (50,50,128) | (50,50,128) | (50,50,8) | (50,50,8) | (100,100,16) |
| **mIoU ↑** | 27.4 | 66.4 | 65.9 | 72.9 | 73.7 | **92.4** |
| **IoU ↑** | 37.0 | 62.3 | 57.7 | 64.1 | 65.1 | **85.6** |

Table 2: **Comparisons of 3D occupancy generation.** We report Inception Score (IS), Fréchet Inception Distance (FID), Kernel Inception Distance (KID), Precision (P), Recall (R), and F-Score (F) in both the **2D** and **3D** domains. † denotes unconditioned generation, while other methods are evaluated using layout conditions. All methods are implemented using official codes and checkpoints.

| Method | #Classes | Metric 2D | | | | | | Metric 3D | | | | | |
|---|---|---|---|---|---|---|---|---|---|---|---|---|---|
| | | IS 2D↑ | FID 2D↓ | KID 2D↓ | P 2D↑ | R 2D↑ | F 2D↑ | IS 3D↑ | FID 3D↓ | KID 3D↓ | P 3D↑ | R 3D↑ | F 3D↑ |
| DynamicCity† [59] | | 1.008 | 7.792 | 8e-3 | 0.108 | 0.009 | 0.017 | 1.269 | 1890 | 0.369 | 0.028 | - | - |
| UniScene [18] | 11 | 1.015 | 0.728 | 5e-4 | 0.295 | 0.572 | 0.389 | 1.278 | 495.6 | 0.027 | 0.387 | 0.482 | 0.429 |
| 𝒳-Scene (Ours) | | **1.030** | **0.275** | **6e-5** | **0.744** | **0.772** | **0.757** | **1.287** | **281.3** | **0.009** | **0.766** | **0.785** | **0.775** |
| UniScene [18] | 17 | 1.023 | 0.770 | 6e-4 | 0.259 | 0.588 | 0.360 | 1.235 | 529.6 | 0.024 | 0.382 | 0.412 | 0.396 |
| 𝒳-Scene (Ours) | | **1.028** | **0.262** | **6e-5** | **0.762** | **0.811** | **0.785** | **1.276** | **258.8** | **0.004** | **0.769** | **0.787** | **0.778** |

tasks, including semantic occupancy prediction (IoU, mIoU), 3D object detection (mAP, NDS), BEV segmentation (mIoU), and end-to-end planning with UniAD (trajectory L2 error and collision rate).

## 4.2 Qualitative Results

**Large-Scale Scene Generation.** The upper part of Figure 5 showcases large-scale scene generation results. By iteratively applying consistency-aware outpainting, 𝒳-Scene effectively expands local regions into coherent, large-scale driving scenes. The generated scenes can be further reconstructed into 3D representations, enabling view rendering and supporting downstream perception tasks. Beyond static environments, our pipeline also produces temporally coherent multi-view videos (see Sec. 4.3 and Fig. 7 for qualitative and quantitative results).

**User-Prompted Generation and Editing.** The lower part of Figure 5 demonstrates the flexibility of 𝒳-Scene in interactive scene generation, supporting both user-prompted generation and geometric editing. Users can provide high-level prompts (e.g., "create a busy intersection"), which are processed to generate corresponding layouts and scene content. Furthermore, given an existing scene, users can specify editing intents (e.g., "remove the parked car") or adjust low-level geometric attributes. Our pipeline updates the scene graph accordingly and regenerates the scene through conditional diffusion.

Table 3: **Comparisons of multi-view image generation**. We report FID and evaluate generation fidelity by performing BEV segmentation [95] and 3D object detection [96] tasks on the generated data from the validation set. **Bold** indicates the best, and underline denotes the second-best results.

| Method | Avenue | Synthesis Resolution | FID↓ | BEV Segmentation | | 3D Object Detection | |
|---|---|---|---|---|---|---|---|
| | | | | Road mIoU ↑ | Vehicle mIoU ↑ | mAP ↑ | NDS ↑ |
| **Original nuScenes** [93] | - | - | - | 73.67 | 34.82 | 35.54 | 41.21 |
| **BEVGen** [36] | RA-L'24 | 224×400 | 25.54 | 50.20 | 5.89 | - | - |
| **BEVControl** [37] | arXiv'23 | - | 24.85 | 60.80 | 26.80 | - | - |
| **DriveDreamer** [3] | ECCV'24 | 256×448 | 26.80 | - | - | - | - |
| **MagicDrive** [1] | ICLR'24 | 224×400 | 16.20 | 61.05 | 27.01 | 12.30 | 23.32 |
| **Panacea** [5] | CVPR'24 | 256×512 | 16.96 | 55.78 | 22.74 | 11.58 | 22.31 |
| **Drive-WM** [19] | CVPR'24 | 192×384 | 15.80 | 65.07 | 27.19 | - | - |
| **DreamForge** [23] | arXiv'25 | 224×400 | 14.61 | 65.27 | 28.36 | 13.01 | 22.16 |
| **Glad** [14] | ICLR'25 | 256×512 | 12.57 | - | - | - | - |
| $\mathcal{X}$-*Scene* (Ours) | - | 224×400 | **11.29** | 66.48 | 29.76 | 16.28 | 26.26 |
| $\mathcal{X}$-*Scene* (Ours) | - | 336×600 | 12.83 | 68.66 | 32.67 | 24.92 | 32.48 |
| $\mathcal{X}$-*Scene* (Ours) | - | 448×800 | 12.77 | **69.06** | **33.27** | **27.65** | **34.48** |

Table 4: **Comparison of multi-view video generation.** We report FVD and assess generation fidelity by evaluating end-to-end planning performance using UniAD [96] on the generated validation data.

| Data Source | Synthesis Resolution | FVD↓ | 3DOD | | BEV Segmentation mIoU (%) | | | | L2 (m) ↓ | | | | Col. Rate (%) ↓ | | | |
|---|---|---|---|---|---|---|---|---|---|---|---|---|---|---|---|---|
| | | | mAP ↑ | NDS ↑ | Lanes↑ | Drivable↑ | Divider↑ | Crossing↑ | 1.0s | 2.0s | 3.0s | Avg. | 1.0s | 2.0s | 3.0s | Avg. |
| Ori nuScenes | 224 × 400 | - | 31.20 | 45.22 | 29.19 | 65.83 | 23.51 | 12.99 | 0.60 | 1.10 | 1.85 | 1.18 | 0.08 | 0.28 | 0.66 | 0.34 |
| MagicDrive [1] | 224 × 400 | 217.9 | 12.92 | 28.36 | 21.95 | 51.46 | 17.10 | 5.25 | 0.57 | 1.14 | 1.95 | 1.22 | 0.10 | 0.25 | 0.70 | 0.35 |
| DreamForge [23] | 224 × 400 | 209.9 | 16.63 | 30.57 | 26.16 | 58.98 | 20.22 | 8.83 | 0.55 | 1.08 | 1.85 | 1.16 | 0.08 | 0.27 | 0.81 | 0.39 |
| $\mathcal{X}$-*Scene* (Ours) | 224 × 400 | **179.7** | **20.40** | **31.76** | **28.04** | **61.96** | **22.32** | **10.48** | **0.55** | **1.08** | **1.81** | **1.15** | **0.03** | **0.13** | **0.66** | **0.27** |

## 4.3 Main Result Comparisons

**Occupancy Reconstruction and Generation.** Table 1 presents the comparative occupancy reconstruction results. The results show that $\mathcal{X}$-*Scene* achieves superior reconstruction performance, significantly outperforming prior approaches under similar compression settings (e.g., +0.8% mIoU and +2.5% IoU compared to UniScene [18]). This improvement is attributed to the enhanced capacity of our triplane representation to preserve geometric details while maintaining encoding efficiency.

Table 2 presents the quantitative results for 3D occupancy generation. Following the protocol in [59], we report performance under two settings: (1) a label-mapped setting, where 11 classes are evaluated by merging similar categories (e.g., car, bus, truck) into a unified "vehicle" class, and (2) the full 17-class setting without label merging. Our approach consistently achieves the best performance across both 2D and 3D metrics. Notably, in the 17-class setting without label mapping, we observe substantial improvements, with FID$^{3D}$ reduced by 51.2% (258.8 vs. 529.6), highlighting our method's capacity for fine-grained category distinction. Additionally, our method demonstrates strong precision and recall, reflecting its ability to generate diverse yet semantically consistent occupancy.

**Image Generation Fidelity.** Table 3 presents the results of multi-view image generation, including FID scores and downstream task evaluations. Notably, $\mathcal{X}$-*Scene* supports high-resolution image generation with competitive fidelity, which is crucial for downstream tasks like 3D reconstruction. The results show that $\mathcal{X}$-*Scene* achieves the best FID, with a 4.91% improvement over the baseline [1], indicating superior visual realism. Moreover, $\mathcal{X}$-*Scene* consistently outperforms other methods in BEV segmentation and 3D object detection as resolution increases. For BEV segmentation in particular, performance on generated scenes at 448×800 resolution closely matches that on real data, showcasing $\mathcal{X}$-*Scene*'s strong conditional generation aligned with downstream visual applications.

**Video Generation Fidelity.** Table 4 presents the results of dynamic video generation and end-to-end evaluation. $\mathcal{X}$-*Scene* is trained on short 7-frame clips using an autoregressive temporal modeling strategy. It achieves a lower FVD than the 16-frame-trained baseline MagicDrive, indicating stronger temporal consistency and video realism with higher efficiency. When evaluated on downstream perception and planning tasks using UniAD, $\mathcal{X}$-*Scene* consistently outperforms the baseline across all metrics. These results demonstrate that $\mathcal{X}$-*Scene* generates temporally coherent and physically consistent dynamic scenes, effectively supporting realistic end-to-end simulation.

Table 5: **Comparisons of training support** for semantic occupancy prediction (Baseline as CONet [97]).

| Data Source | Input Modality | 3D Occ Pred. IoU ↑ | mIoU ↑ |
|---|---|---|---|
| Ori nuScenes | 2D (Images) | 20.1 | 12.8 |
| +MagicDrive [1] | | 21.8 | 13.9 |
| +UniScene [18] | | 28.6 | 16.5 |
| +$\mathcal{X}$-Scene (Ours) | | **29.1** | **17.2** |
| Ori nuScenes | 3D (LiDAR/Occ) | 30.9 | 15.8 |
| +UniScene [18] | | 33.1 | 19.3 |
| +$\mathcal{X}$-Scene (Ours) | | **35.8** | **22.6** |
| Ori nuScenes | 2D+3D | 29.5 | 20.1 |
| +UniScene [18] | | 35.4 | 23.9 |
| +$\mathcal{X}$-Scene (Ours) | | **37.1** | **26.3** |

Table 6: **Comparison of training support** for BEV segmentation (Baseline as CVT [95]) and 3D object detection (Baseline as StreamPETR [98] following the setup in [23, 5]).

| Data Type | Data Source | 3D Object Detection mAP ↑ | NDS ↑ | mAoE ↓ | BEV Segmentation Rd. mIoU ↑ | Veh. mIoU ↑ |
|---|---|---|---|---|---|---|
| **Real** | Ori nuScenes | 34.5 | 46.9 | 59.4 | 74.30 | 36.00 |
| **Gen.** | Panacea [5] | 22.5 | 36.1 | 72.7 | - | - |
| | DreamForge [23] | 26.0 | 41.1 | 62.2 | 67.80-6.50 | 28.60-7.40 |
| | $\mathcal{X}$-Scene (Ours) | **28.2** | **43.4** | **61.0** | **68.41**-5.89 | **29.23**-6.77 |
| **Real+Gen** | Vista [2] | 34.0 | 38.6 | - | 76.62+2.32 | 37.71+1.71 |
| | MagicDrive [1] | 35.4 | 39.8 | - | 79.56+5.26 | 40.34+4.34 |
| | UniScene [18] | 36.5 | 41.2 | - | 81.69+7.39 | 41.62+5.62 |
| | DreamForge [23] | 36.6 | 49.5 | 52.9 | - | - |
| | Panacea [5] | 37.1 | 49.2 | 54.2 | - | - |
| | $\mathcal{X}$-Scene (Ours) | **39.9** | **51.6** | **51.2** | **83.37**+9.07 | **43.05**+7.05 |

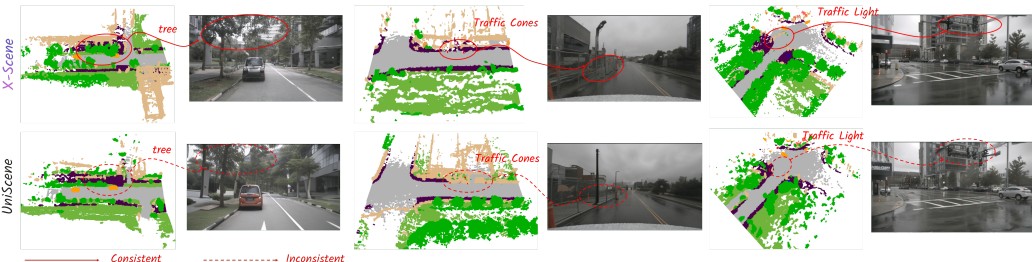

Figure 6: **Qualitative comparison of joint voxel-and-image generation**. Our method achieves superior consistency between generated 3D occupancy and 2D images compared to UniScene [18].

**Downstream Tasks Evaluation.** We evaluate the effectiveness of the generated scene data in supporting downstream model training. Table 5 reports the results for 3D semantic occupancy prediction. Fine-tuning with our synthesized 3D occupancy grids notably improves the baseline performance (+4.9% IoU, +6.8% mIoU), as the high-resolution grids provide accurate and detailed spatial structures that enable better geometric reasoning and feature learning. Moreover, integrating 2D and 3D modalities yields the highest performance, demonstrating the importance of multimodal alignment. Table 6 presents the results for 3D object detection and BEV segmentation. Our generated data consistently surpasses all synthetic data baselines, verifying the superior fidelity, realism, and temporal consistency of our approach. Overall, these results confirm the potential of our synthesized images and videos to serve as high-quality data augmentation for downstream models.

**Qualitative Comparisons.** Figure 6 illustrates a comparison of joint voxel-and-image generation results. $\mathcal{X}$-Scene produces more realistic and diverse scenes while maintaining tighter geometric alignment between 3D occupancy and 2D images, leading to improved cross-modal coherence. Figure 7 further showcases qualitative results of multi-view video generation. $\mathcal{X}$-Scene generates temporally coherent sequences with smoother motion transitions and stable object dynamics, while maintaining accurate cross-view geometry and visual consistency. Together, these results demonstrate $\mathcal{X}$-Scene's ability to generate spatially coherent 3D structures and photorealistic, temporally consistent videos, offering a scalable and reliable foundation for simulation and data generation.

### 4.4 Ablation Study

**Effects of Designs in Occupancy Generation.** As shown in Table 7, the proposed triplane deformable attention module improves performance, particularly at lower resolutions. For instance, at a (50, 50, 16) resolution, introducing deformable attention yields gains of +1.9% IoU and +2.4% mIoU, confirming its role in alleviating feature degradation caused by downsampling. We further examine the impact of conditioning strategies. Removing either the additive layout condition or the box condition leads to noticeable performance drops, highlighting their complementary contributions. These conditions provide essential fine-grained geometric cues that guide the model to better capture scene structure and spatial context, ultimately improving occupancy field accuracy.

Table 7: **Ablation study** for designs in the occupancy generation model.

| Variants | Triplane Resolution | IoU↑ | mIoU↑ | FID$^{3D}$↓ | F$^{3D}$↑ |
|---|---|---|---|---|---|
| $\mathcal{X}$-Scene (Ours) | (100,100,16) | **85.6** | **92.4** | **258.8** | **0.778** |
| w/ VAE deform attn | (50,50,16) | 66.6 | 76.6 | 436.1 | 0.522 |
| w/o VAE deform attn | (50,50,16) | 64.7 | 74.2 | 462.4 | 0.510 |
| w/o VAE deform attn | (100,100,16) | 84.9 | 91.8 | 266.4 | 0.762 |
| w/o layout Condition | (100,100,16) | 85.6 | 92.4 | 1584 | 0.237 |
| w/o box Condition | (100,100,16) | 85.6 | 92.4 | 271.4 | 0.751 |

Table 8: **Ablation study** for designs in the multi-view image generation model.

| Variants | FID | 3D Detection | | BEV Segmentation | |
|---|---|---|---|---|---|
| | | mAP ↑ | NDS ↑ | Rd. mIoU ↑ | Veh. mIoU ↑ |
| $\mathcal{X}$-Scene (Ours) | **11.29** | **16.12** | **26.26** | **66.48** | **29.60** |
| w/o semantic map | 12.23 | 15.27 | 25.59 | 65.75$_{-0.73}$ | 28.71$_{-0.89}$ |
| w/o depth map | 12.94 | 15.61 | 25.98 | 64.87$_{-1.61}$ | 29.22$_{-0.38}$ |
| w/o perspective map | 16.87 | 13.15 | 22.37 | 63.35$_{-3.13}$ | 27.13$_{-2.47}$ |
| w/o position embed | 11.38 | 15.60 | 26.16 | 66.46$_{-0.02}$ | 27.88$_{-1.72}$ |
| w/o text description | 12.60 | 15.54 | 26.06 | 66.26$_{-0.22}$ | 29.47$_{-0.13}$ |

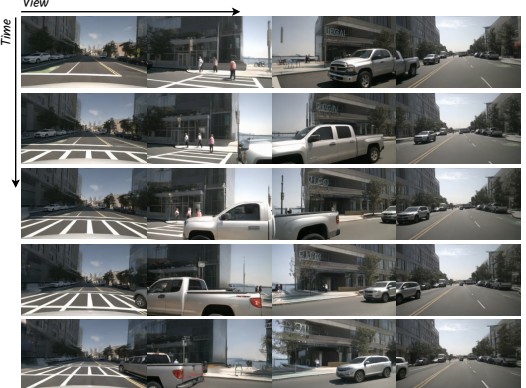
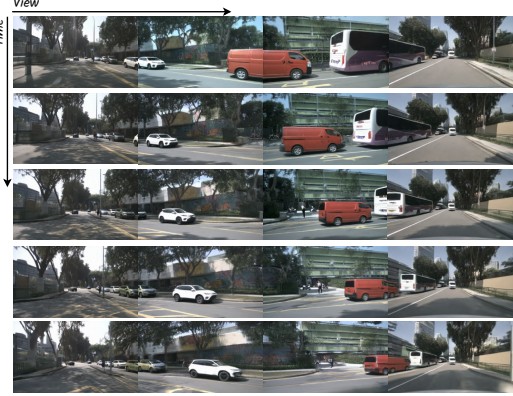

Figure 7: **Qualitative comparison of multi-view video generation**. Our method demonstrates superior temporal consistency across frames and spatial coherence among multiple camera views.

**Effects of Designs in Image Generation.** Table 8 presents the ablation results for various conditioning components in the image generation model. Removing the semantic or depth maps that are rendered from 3D occupancy significantly degrades FID and downstream performance, highlighting their importance in providing dense geometric and semantic cues. Excluding the perspective map, which encodes projected 3D boxes and lanes, also reduces downstream performance (with mAP dropping by 2.97%), underscoring its role in conveying explicit layout priors. The 3D positional embedding is particularly critical for object detection, as it enhances localization and spatial representation. Finally, removing the text description degrades generation fidelity (FID worsening by 1.31%), showing that rich linguistic context aids fine-grained appearance modeling and scene understanding.

## 5 Conclusion and Limitations

In this paper, we present $\mathcal{X}$-*Scene*, a novel framework for 3D driving scene generation that achieves high fidelity, flexible controllability, and large-scale spatial and temporal consistency. Leveraging the multi-granular control mechanism, $\mathcal{X}$-*Scene* allows intuitive yet precise specification of both high-level semantic guidance and low-level geometric details. Its unified voxel–image–video generation pipeline captures detailed 3D geometry, photorealistic appearance, and temporally coherent dynamics, while consistency-aware outpainting maintains spatial coherence across expansive environments. Extensive experiments show that $\mathcal{X}$-*Scene* outperforms existing approaches in generation quality, controllability, and scalability, establishing it as a versatile tool for large-scale data generation, driving simulation, and interactive scene exploration. Future work will explore longer temporal horizons and multi-agent interactions to further enhance the realism and dynamism of generated driving scenarios.

## 6 Acknowledgments

This research was supported by the Tier 2 Grant (MOE-T2EP20124-0015) from the Singapore Ministry of Education and by the National Natural Science Foundation of China (Grant No. 62525309).

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

# $\mathcal{X}$-Scene: Large-Scale Driving Scene Generation with High Fidelity and Flexible Controllability

## Supplementary Material

## Contents

# A  Additional Implementation Details

In this section, we provide additional implementation details to facilitate reproducibility. Specifically, we elaborate on the experimental datasets, model implementation, and the evaluation metrics.

## A.1  Datasets

We use Occ3D-nuScenes [92] to train our controllable occupancy generation module, and nuScenes [93] for the multi-view image and video generation modules. The textual scene graph-to-layout generation module is also trained using 3D bounding box and HD map annotations from nuScenes. The dataset comprises 1,000 driving scenes under diverse weather, lighting, and traffic conditions. Each 20-second scene includes about 40 annotated keyframes, yielding roughly 40,000 samples with 360° multi-view images, 3D occupancy, bounding boxes, and maps. We follow the standard split of 700 training and 150 validation scenes. For video generation, ASAP interpolation is applied to upsample the frame rate from 2 Hz to 12 Hz, yielding about 240 frames per scene and enabling more consistent training for temporally coherent video synthesis. Following DynamicCity [59], we map the original 17 semantic categories to 11 commonly used classes (see Table 9) and conduct experiments both with and without label mapping to enable comprehensive comparisons.

Table 9: **Summary of Semantic Label Mappings**. We map the original 17-class nuScenes semantic labels to 11 classes following the protocol in [59] to enable comprehensive evaluation.

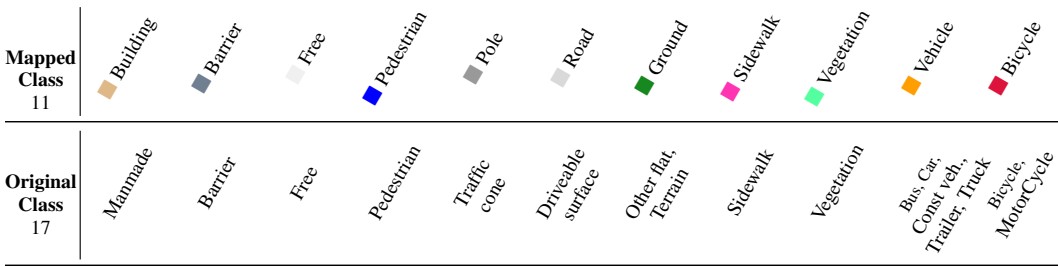

## A.2  Model Implementation Details

**Textual Scene Description Generation Module.**  To construct the scene description memory bank $\mathcal{M}$, we utilize QWen2.5-VL [99] to extract structured information from nuScenes. For each frame, six surround-view images are jointly processed to generate holistic scene descriptions, which are parsed into scene style $\mathcal{S}$, foreground objects $\mathcal{O}$, and background elements $\mathcal{B}$. Concurrently, 3D bounding boxes and lane markings are converted into textual scene-graph layouts $\mathcal{L}$. These components collectively form memory entries $m_i = \{\mathcal{S}, \mathcal{O}, \mathcal{B}, \mathcal{L}\}$.

For retrieval, text descriptions are encoded using OpenAI's text-embedding-3-small model and indexed with FAISS to enable efficient similarity search. During inference, given a coarse prompt $\mathcal{T}_{\mathcal{P}}$, we retrieve the top-$K$ relevant entries from $\mathcal{M}$, which are then combined with the prompt and fed into GPT-4o to generate a detailed and structured scene description $\mathcal{D}$. Please refer to Sec. B for further details and example illustrations.

**Scene-Graph to Layout Generation Module.**  For the scene-graph to layout generation module, training and evaluation were conducted on a single NVIDIA A6000 GPU with 48GB of memory. We employed a batch size of 128 and trained the model for 400 epochs. The optimization was performed using the AdamW optimizer with an initial learning rate of $1 \times 10^{-4}$ and a cosine annealing scheduler. To ensure stable training and consistent representation, the 3D bounding boxes were normalized using dataset-specific parameters. Each bounding box $b_i$ was parameterized by its center coordinates $(x, y, z)$, dimensions $(l, w, h)$, and yaw angle $\theta$. Following standard practices in 3D object detection, we normalized the box center coordinates to the range $[0, 1]$, applied a logarithmic transformation to the dimensions, and represented the yaw angle using its sine and cosine components. Each graph node was augmented with an 8-dimensional noise vector to enhance robustness during training.

**Occupancy Generation Module.** For the occupancy generation module, the triplane-VAE encodes the original occupancy field with a resolution of $200 \times 200 \times 16$ into a triplane representation of spatial dimensions $(X_h, Y_h, Z_h) = (100, 100, 16)$ and feature dimension $C_h = 16$, reducing memory consumption while preserving structural details. The triplane-VAE is trained using the Adam optimizer with an initial learning rate of $1 \times 10^{-3}$ and a step decay factor of 0.1, over 200 epochs on 4 NVIDIA A6000 GPUs with a batch size of 24 per GPU.

During diffusion, the three orthogonal planes are arranged into a unified square feature map by zero-padding the uncovered corners, forming a tensor $\mathbf{h} \in \mathbb{R}^{X_h+Z_h, Y_h+Z_h, C_h}$. Attention is applied across this tensor to capture inter-plane correlations. The diffusion model is trained from scratch using the AdamW optimizer with an initial learning rate of $1 \times 10^{-4}$ and a cosine scheduler, over 300 epochs with a batch size of 12 per GPU. For occupancy outpainting, we adopt the RePaint sampling strategy with 5 resampling steps and a jump size of 20.

**Multi-View Image Generation Module.** We initialize the multi-view image generation module with pretrained Stable Diffusion v2.1 weights, while randomly initializing newly added parameters. The diffusion model is trained on 4 NVIDIA A6000 GPUs with a mini-batch size of 8, using the AdamW optimizer with a learning rate of $8 \times 10^{-5}$ and a cosine learning rate scheduler over 200 epochs. After initial training at a resolution of $224 \times 400$, we fine-tune the model for an additional 50K iterations at higher resolutions of $448 \times 800$ and $336 \times 600$. During inference, we use the UniPC [100] scheduler with 20 steps and a Classifier-Free Guidance (CFG) scale of 1.2.

**Multi-View Video Generation Module.** We initialize the multi-view video generation module using the pretrained image diffusion U-Net and focus on fine-tuning the newly introduced temporal attention layers. The training is performed for 100 epochs with a total batch size of 8, where two reference frames are randomly sampled from the preceding five ground-truth frames, and each training sample contains 7 frames in total. For higher-resolution settings, we further train the model for 50K iterations, initializing from the corresponding lower-resolution weights. The temporal module is trained on eight NVIDIA A100 GPUs using the AdamW optimizer with a learning rate of $8 \times 10^{-5}$ and a cosine learning rate scheduler.

During inference, the reference frames are drawn from previously generated video clips. For the first clip, we employ the single-frame image generation model to produce the initial reference frame, after which the system follows an autoregressive generation strategy. By default, two reference frames are used to generate the subsequent seven frames, enabling temporally coherent and geometrically consistent video synthesis across multiple views.

## A.3 Evaluation Metrics for Occupancy Generation

Following the evaluation protocol of DynamicCity [59], we adopt two complementary strategies to assess the quality of occupancy generation:

• **3D Evaluation**: We train a sparse convolutional autoencoder based on the MinkowskiUNet [101] architecture to extract 3D features from generated occupancy fields. Features from the final down-sampling layer are aggregated via global average pooling and used to compute evaluation metrics using the Torch-Fidelity library [102].

• **2D Evaluation**: We render the 3D occupancy fields into 2D images for image-based evaluation. To ensure fair comparison, we standardize the rendering process across all methods using consistent semantic color mappings and camera parameters. We compute IS, FID, and KID using a standard pretrained InceptionV3 [103] network, and use a VGG-16 [104] model for precision and recall. Both networks are fine-tuned on our semantically color-mapped dataset to ensure domain alignment.

To evaluate the quality and diversity of the generated samples, we use several quantitative metrics: **1) Inception Score (IS)** measures both quality and diversity via the KL divergence between each image's conditional label distribution and the marginal distribution, with higher scores indicating sharper and more diverse samples; **2) Fréchet Inception Distance (FID)** computes the distance between real and generated distributions in the Inception feature space, where lower values indicate higher fidelity; **3) Kernel Inception Distance (KID)** calculates the squared Maximum Mean Discrepancy (MMD) between real and generated features using a polynomial kernel, and is unbiased and less sensitive to sample size; **4) Precision** estimates the proportion of generated samples within the support of real

**Algorithm 1:** Textual Scene Description Generation via VLM, LLM, and RAG

---

**Input:** User prompt $\mathcal{T}_{\mathcal{P}}$; Scene dataset $\mathcal{D}_{\text{scene}}$
**Output:** Structured scene description $\mathcal{D} = \{\mathcal{S}, \mathcal{O}, \mathcal{B}, \mathcal{L}\}$

**1** **Offline Stage: Build Memory Bank** $\mathcal{M}$
**2** **for** *frame $f$ in $\mathcal{D}_{scene}$* **do**
**3**     Load 6 surround-view images $I_f$ ;
**4**     $\hat{d}_f \leftarrow \texttt{VLM}(I_f)$ ;             // Generate raw description
**5**     $\mathcal{S}, \mathcal{O}, \mathcal{B} \leftarrow \texttt{Parse}(\hat{d}_f)$ ;      // Parse style, objects, and background
**6**     $A_f \leftarrow \texttt{DataAnnotations}(f)$ ;     // Extract spatial annotations
**7**     $\mathcal{L} \leftarrow \texttt{LayoutFrom}(A_f)$ ;     // Convert annotations to textual layout
**8**     $m_f \leftarrow \{\mathcal{S}, \mathcal{O}, \mathcal{B}, \mathcal{L}, \hat{d}_f\}$ ;     // Assemble memory item
**9**     Add $m_f$ to memory bank $\mathcal{M}$ ;

**10** **Online Stage: Generate Structured Description** $\mathcal{D}$
**11** $z_{\mathcal{P}} \leftarrow \texttt{Embed}(\mathcal{T}_{\mathcal{P}})$ ;          // Embed user prompt
**12** $\{z_i\} \leftarrow \texttt{Embed}(m_i.\text{text})$ for all $m_i \in \mathcal{M}$ ;   // Embed memory entries
**13** $\mathcal{M}_K \leftarrow \texttt{TopK}(z_{\mathcal{P}}, \{z_i\})$ ;    // Retrieve top-k relevant memories with RAG
**14** Format LLM input using $\mathcal{T}_{\mathcal{P}}$ and $\mathcal{M}_K$ ;     // Prepare input context
**15** $\mathcal{D} \leftarrow \mathcal{G}_{\text{description}}(\mathcal{T}_{\mathcal{P}}, \mathcal{M}_K)$ ;    // Generate final description via GPT-4o

---

data; **5) Recall** measures how well the generated distribution covers real data; and **6) F1-Score**, the harmonic mean of precision and recall, reflects the balance between generation quality and coverage.

## B   Additional Details of Scene Description Generation

The scene description module constructs textual scene representations by integrating vision-language models (VLMs) and large language models (LLMs). As shown in Algorithm 1, a scene memory bank is first built offline using a VLM. During inference, a RAG pipeline selects the most relevant memory items based on a user's coarse prompt, enabling the LLM to generate detailed, context-grounded scene descriptions. This framework supports flexible and scalable scene description generation.

### B.1   Scene Description Memory Construction

To construct the scene description memory bank $\mathcal{M}$, we use QWen2.5-VL [99] to extract structured scene information from the nuScenes dataset. For each annotated timestamp, the six surround-view camera images are processed by the VLM to generate a holistic natural language description, which is parsed into structured components $\{\mathcal{S}, \mathcal{O}, \mathcal{B}\}$: scene style (e.g., "a rainy afternoon in an urban area"), foreground objects with spatial and appearance attributes (e.g., "a red sedan parked alongside the walkway"), and background elements (e.g., "high-rise buildings in the distance"). In parallel, nuScenes 3D bounding boxes and lane markings are converted into a textual scene-graph layout $\mathcal{L}$ capturing spatial relationships (e.g., "car A is behind truck B", "pedestrian is on the sidewalk near lane L1"). Together, these components form each memory item $m_i$.

### B.2   Novel Scene Description Generation

During inference, given a coarse user prompt $\mathcal{T}_{\mathcal{P}}$, we employ GPT-4o as the LLM-based generator $\mathcal{G}_{\text{description}}$ and implement a RAG mechanism to enrich the prompt with relevant memories. Specifically, both the prompt and the entries in the memory bank $\mathcal{M}$ are embedded using a pre-trained sentence embedding model (i.e., text-embedding-3-small). We then retrieve the top-K most semantically similar descriptions from $\mathcal{M}$. These retrieved examples serve as contextual references, enabling the LLM to generate a rich and coherent scene description $\mathcal{D} = \{\mathcal{S}, \mathcal{O}, \mathcal{B}, \mathcal{L}\}$ tailored to the user's input.

This RAG design is motivated by the need to bridge coarse user prompts and fine-grained scene representations, enabling few-shot generalization and knowledge transfer from similar scenes in the memory bank. Furthermore, the memory bank $\mathcal{M}$ is modular and extensible, supporting future inclusion of other datasets with minimal adaptation effort.

## B.3 Prompt Details and Scene Description Examples

The following system prompt is defined for **constructing scene description memories**. Given two images capturing the 360-degree surroundings, the VLM is guided to **extract and organize key elements of the driving scene** into a comprehensive representation:

---

**System prompt for scene description memory construction with VLM**

Given two panoramic images **<image>FRONT_IMAGE</image>** and **<image>BACK_IMAGE</image>** that encapsulate the surroundings of a vehicle in a 360-degree view, your task is to analyze the driving scene.
Your analysis should include the following core information:

- **Time of the day**: Indicate whether it is daytime or nighttime.
- **Weather**: Specify if it is sunny, rainy, cloudy, snowy, or foggy.
- **Surrounding environment**: Classify the environment as downtown, urban expressway, suburban, rural, highway, residential, industrial, nature, etc.
- **Foreground objects**: Identify objects in the foreground, such as cars, buses, trucks, pedestrians, bicycles, motorcycles, construction vehicles, barriers, traffic cones, traffic signs/lights, etc.
- **Background elements**: Describe background elements, including roads, sidewalks, pedestrian crossings, car parks, terrain, vegetation, buildings, etc.
- **Road condition**: Characterize the road as an intersection, straight road, narrow street, wide road, pedestrian crossing, etc.
- **Abstract Description**: Provide a concise summary of the scene, integrating details about scene features, foreground objects, background information, and road conditions.

**Instruction:**

- Each panoramic image consists of three smaller images. The first image covers the left-front, directly in front, and right-front views of the vehicle. The second image includes the left-rear, directly behind, and right-rear views.
- When describing **foreground objects**, clearly detail their unique appearance and location. Specify each object's relative position to the ego vehicle using terms like front, back, left, right, etc. Avoid referencing terms like *"first/second image"* or directional phrases such as *"front-left/rear-center view"*. If there are multiple objects of the same type, provide a description for each one.
- For **background elements**, provide descriptions of their notable characteristics.
- Assess the presence of objects from the provided candidate list. If an object exists, describe its attributes briefly. If it does not exist, omit it from your output. You may also include objects not listed in the candidates.

**Please format your results as follows:**

```
{
  "Time of the day": "xxx",
  "Weather": "xxx",
  "Surrounding environment": "xxx",
  "Foreground objects": [
    {"object1 class": "object1 attributes"},
    ...
  ],
  "Background elements": [
    {"element1 class": "element1 attributes"},
    ...
  ],
  "Road condition": "xxx",
  "Abstract Description": "xxx"
}
```

**Example output:**

```
{
  "time of the day": "daytime",
  "Weather": "sunny",
  "Surrounding environment": "downtown",
  "Foreground objects": [
    {"car": "blue color"},
    {"truck": "gray color"},
  ],
  "Background elements": [
    {"sidewalk": "narrow"},
    {"building": "white color and tall"}
  ],
  "Road condition": "intersection",
  "Abstract Description": "The scene depicts a sunny day in a downtown area with a
      blue car, gray truck, and a crouching pedestrian. The narrow sidewalk is
      lined with a white, tall building."
}
```

---

The following system prompt is defined for **generating novel scene descriptions**. Given a coarse user prompt, the LLM is guided to retrieve semantically relevant scene descriptions from a structured memory bank. These retrieved references are then used to **enrich, clarify, and ground** the final output, resulting in a coherent and contextually accurate scene description:

---

### System prompt for novel scene description generation with LLM+RAG

You are an **intelligent assistant** for detailed driving scene understanding and generation. Given a **coarse user prompt** and a set of **relevant memory items** retrieved from a structured memory bank, your task is to generate a comprehensive, structured description of the target driving scene.

**Input Tokens:**

- **\<text\>USER_PROMPT\</text\>**: a high-level, possibly ambiguous user query describing a scene, e.g., *"a busy urban street at night"*.

- **\<JSON\>MEMORY\</JSON\>**: a collection of scene descriptions in JSON format, semantically retrieved as relevant references to the prompt.

These memory examples should be used to **enhance**, **clarify**, and **ground** your final output. Your output must strictly follow the specified JSON structure and provide a **cohesive and concrete** description of the driving scene.

**Instructions:**

- Leverage relevant entries in **\<memory\>** to help **expand**, **clarify**, or **disambiguate** the user's **\<prompt\>**.

- When information in the prompt is sparse or vague, **infer plausible details** based on common patterns from similar memory entries.

- Be **specific** in describing the following:
    - Spatial relationships between objects (e.g., beside, ahead, behind)
    - Object attributes (e.g., color, type, behavior)
    - Environmental context (e.g., weather, road type, background elements)

- **Do not directly copy** content from memory items; instead, **synthesize** a new, coherent scene inspired by them.

- **Avoid referencing** the tokens **\<prompt\>** or **\<memory\>** in the output.

---

Expected Output Structure (JSON):

```
{
  "Time of the day": "xxx",
  "Weather": "xxx",
  "Surrounding environment": "xxx",
  "Foreground objects": [
    {"object1": "object1 attributes"},
    ...
  ],
  "Background elements": [
    {"element1": "element1 attributes"},
    ...
  ],
  "Road condition": "xxx",
  "Layout": "xxx",
  "Abstract Description": "xxx"
}
```

---

Example Output:

```
{
  "Time of the day": "daytime",
  "Weather": "cloudy",
  "Surrounding environment": "urban expressway",
  "Foreground objects": [
    {"truck": "red container truck driving ahead"},
    {"motorcycle": "black motorcycle overtaking from the right"}
  ],
  "Background elements": [
    {"overpass": "grey concrete with visible traffic signs"},
    {"vegetation": "small bushes along the road divider"}
  ],
  "Road condition": "straight road",
  "Layout": "The truck is positioned ahead of the motorcycle and is located on the
      drivable road. The motorcycle is adjacent to the pedestrian crossing.",
  "Abstract Description": "A cloudy daytime scene on an urban expressway, with a
      red container truck ahead and a black motorcycle overtaking. Concrete
      overpasses and roadside bushes shape the background."
}
```

Representative examples of the generated scene descriptions, including scene style, foreground objects, background elements, and scene-graph layouts, are presented below:

---

### Scene Description Example

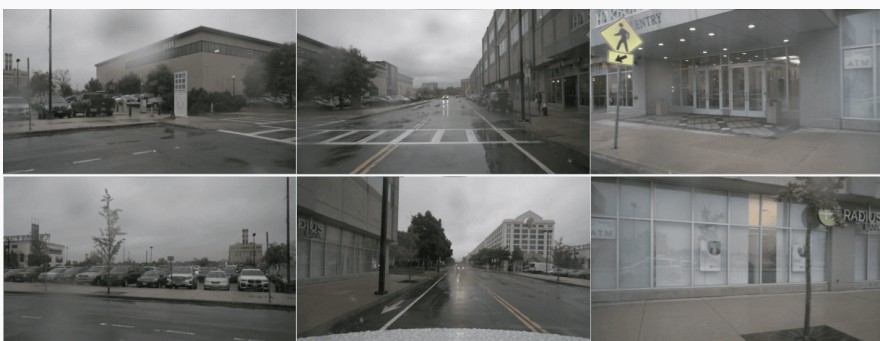

**Scene Style:**
- Time of the day: daytime, with diffused lighting due to cloud cover
- Weather: light to moderate rain, as indicated by raindrops on the camera lens and wet pavement
- Surrounding environment: urban street flanked by mixed-use buildings and parking areas
- Road condition: long, straight, two-lane road with clearly marked crosswalks and lane dividers; visibly slick from rainfall

**Foreground Objects:**
- Cars (left side): a variety of parked vehicles, including a dark pickup truck, compact sedans, SUVs, and so on, aligned parallel along the sidewalk; some cars have reflections on the wet ground
- Cars (right side): several vehicles parked curbside in front of commercial buildings, including economy cars to midsize SUVs
- Pedestrian: an individual wearing a bright orange reflective safety vest and holding a red umbrella, standing near a crosswalk, suggesting a crossing action in progress
- Traffic cone: bright orange cone placed on the sidewalk near the edge of a parking entrance, likely for safety or to reserve space
- Traffic sign: a yellow pedestrian crossing sign mounted on a pole; the sign is positioned close to a glass-door building entrance

**Background Elements:**
- Road: appears dark and glossy due to recent rainfall; lane markings and crosswalks are clearly visible
- Sidewalk: wide, concrete sidewalks run alongside the road, bordered by planters and lined with trees
- Buildings: prominent structures include multi-level modern commercial buildings with large glass façades
- Trees: scattered urban landscaping includes small trees planted at regular intervals, offering a touch of greenery
- Car Parking: multiple designated parking areas, some directly along the street and others within enclosed lots
- Crosswalk: wide white-striped pedestrian crossings present at intersections
- Streetlights: installed at intervals along the road to provide visibility during low-light conditions

**Scene-Graph Layout:**

**Traffic Light Existing**: False
├── **Crosswalk** [(-9.6, -50.1), (-9.4, -43.3), (-12.8, -43.2), (-13.0, -50.0)]
├── Current **straight lane** [(-0.7, -15.0), (-0.5, -8.2), (-0.3, -1.4), (-0.1, +5.4)]
│   ├── ego vehicle **on top of** the lane
├── **Straight lane** with-flow [(-1.5, -45.0), (-1.3, -39.2), (-1.2, -33.4), (-1.0, -27.6)]
│   ├── **vehicle.car on top of** the lane, same direction as ego in the **left back**, location: (-1.8, -32.1, +1.1).
├── **Straight lane** allowing from left to right driving [(+6.3, -20.6), (+12.5, -20.5), (+18.7, -20.5), (+24.9, -20.5)]
├── **Straight lane** opposite-flow [(-4.8, -37.9), (-4.7, -41.9), (-5.3, -45.9), (-4.8, -49.9)]
├── Other Lanes/Roadside
│   ├── movable_object.**trafficcone** in the **left front** location: (-11.4, +9.9, +0.3).length: 0.9, width: 0.4, height: 0.5.
│   ├── movable_object.**trafficcone** in the left front location: (-13.4, +10.0, +0.4).length: 0.4, width: 0.3, height: 0.8.
│   ├── **vehicle.car**.parked in the **left**, heading from left to right, location: (-18.6, -0.1, +0.6).length: 4.2, width: 1.9, height: 1.4.
│   ├── **vehicle.car**.parked in the **left back**, heading from left to right, location: (-18.6, -2.6, +0.9).length: 4.6, width: 2.1, height: 1.8.
│   ├── **vehicle.car**.parked in the **left back**, heading from left to right, location: (-18.4, -5.2, +0.6).length: 4.2, width: 1.9, height: 1.5.
│   ├── **human.pedestrian**.moving in the **right front**, opposite direction, location: (+7.2, +20.0, +1.2).length: 0.9, width: 0.7, height: 1.7.
│   ├── **human.pedestrian**.standing in the **left back**, heading from left to right, location: (-13.2, -30.6, +0.8).length: 0.6, width: 0.6, height: 1.7.
│   ├── **human.pedestrian**.standing in the **left back**, heading from left to right, location: (-13.2, -31.3, +0.9).length: 0.5, width: 0.6, height: 1.7.
│   ├── ......

**Abstract Description:**
- The scene shows a rainy day on an urban expressway with wet roads reflecting light. Numerous cars are parked on both sides of the street, and a pedestrian in a bright orange safety vest is near the crosswalk. The background features multi-story commercial buildings with large windows, trees lining the sidewalk, and various signage. Streetlights are visible, and the area is marked with crosswalks and lane lines, creating a realistic and structured urban driving scene.

## Scene Description Example

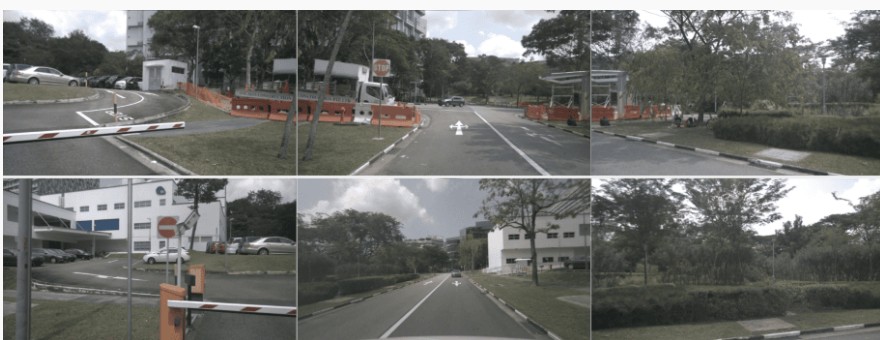

**Scene Style:**
- Time of the day: bright daytime with clear shadows, indicating direct sunlight and good visibility
- Weather: sunny with partly cloudy skies; no signs of precipitation or poor visibility
- Surrounding environment: suburban campus or business park-like area with a mix of roadways, pedestrian paths, and landscaped
- Road condition: smooth asphalt with a gentle curve transitioning into a straight segment

**Foreground Objects:**
- Barriers (construction zone): bright orange modular barriers clearly indicating a temporarily restricted area
- Truck (construction zone): a white construction truck with visible text parked adjacent to the barriers
- Stop sign (construction zone): standard red octagonal stop sign mounted on a metallic pole, reinforcing right-of-way rules
- Entrance barrier: red and white striped boom barrier at two vehicle access points, indicating controlled entry
- Cars: silver sedan (parked at curve near the booth); black sedan (alongside silver); dark-colored sedan (moving towards camera, on central lane); red and silver vehicles (visible in side/rear view near building and behind the no-entry sign)
- Seated pedestrian: a person is resting on the grassy area near the right side of the road, shaded by trees

**Background Elements:**
- Road: dual-lane road with center markings and white directional arrows
- Sidewalk: paved pathways flanked by grass and shrubs on both sides of the road
- Buildings: main visible structure is a multi-story white facility with large windows and blue signage
- Vegetation: tall green trees forming canopy along both sides of the road, sculpted bushes reinforce the planned landscape design
- Car Parking: clearly delineated lot visible on the left, populated with multiple parked cars
- Traffic Signage: a "No Entry" (red circle with horizontal white bar) sign prominently displayed at an access control gate

**Scene-Graph Layout:**

**Traffic Light Existing**: False
├── Current **straight lane** [(-0.2, -0.6), (-0.1, +2.0), (0.0, +4.6), (+0.1, +7.2)]
│   ├── **ego vehicle on top of** the lane
├── **Straight lane** with-flow [(+0.1, +7.2), (+0.2, +10.5), (+0.3, +13.7), (+0.5, +16.9)]
│   ├── movable_object.**trafficcone** in the **left front** location: (-1.7, +14.3, +0.6).
│   ├── movable_object.**trafficcone** in the **left front** location: (-1.7, +15.0, +0.5).
├── **Right turning lane** allowing from right to left driving [(+5.9, +50.0), (+5.7, +41.4), (-1.4, +34.9), (-9.8, +34.3)]
│   ├── **vehicle.car**.moving in the **right front**, heading from right to left, location: (+0.4, +37.7, +1.3).
├── **Left turning lane** allowing from left to right driving [(-8.8, +44.4), (-5.3, +45.1), (-1.9, +46.9), (+0.1, +50.0)]
│   ├── **vehicle.car**.moving in the **left back**, same direction as ego, location: (-1.2, -29.3, +0.8).
├── Other Lanes/Roadside
│   ├── movable_object.**barrier** in the **left front** location: (-2.7, +12.9, +0.7).length: 0.5, width: 2.3, height: 1.2.
│   ├── movable_object.**barrier** in the **left front** location: (-3.7, +13.2, +0.8).length: 0.4, width: 2.4, height: 1.2.
│   ├── movable_object.**trafficcone** in the **left front** location: (-3.2, +18.8, +0.7).length: 0.3, width: 0.3, height: 0.7.
│   ├── vehicle.**bicycle**.without_rider in the **right front**, opposite direction from ego, location: (+9.7, +9.0, +0.2).length: 1.4, width: 0.4, height: 1.1.
│   ├── vehicle.**bicycle**.without_rider in the **right front**, opposite direction from ego, location: (+10.3, +8.7, +0.2).length: 1.4, width: 0.5, height: 1.2.
│   ├── **vehicle.car**.parked in the **left front**, same direction as ego, location: (-19.3, +4.6, +3.2).length: 4.7, width: 1.9, height: 1.8.
│   ├── **vehicle.truck**.parked in the **left front**, heading from left to right, location: (-6.4, +14.5, +2.0).length: 6.5, width: 2.3, height: 3.3.
│   ├── **human.pedestrian**.adult.sitting_lying_down in the **right front**, heading from right to left, location: (+9.9, +10.9, +0.2).
│   ├── **human.pedestrian**.adult.sitting_lying_down in the **right front**, heading from right to left, location: (+9.5, +11.7, +0.3).
│   ├── ......

**Abstract Description:**
- The scene shows a sunny day in a suburban area with a mix of urban infrastructure and greenery. The road curves slightly before straightening out, with construction barriers and a stop sign indicating ongoing work. Several cars are parked along the roadside, while others are in motion. A pedestrian is seated on the grass near the right-rear view, and there are trees and buildings in the background. The road appears well-maintained, with clear lane markings and a mix of open spaces and developed areas.

Table 10: Ablation on text-only generation.

| Variants | FID↓ | 3DOD | | BEVSeg mIoU (%) | |
| --- | --- | --- | --- | --- | --- |
| | | mAP↑ | NDS↑ | Road↑ | Vehicle↑ |
| Full Model | **11.29** | **16.28** | **26.26** | **66.48** | **29.76** |
| Text Only | 20.74 | 2.13 | 5.34 | 28.32 | 7.49 |

Table 11: Ablation on input layout types.

| Input Layout | FID↓ | 3DOD | | BEVSeg mIoU (%) | |
| --- | --- | --- | --- | --- | --- |
| | | mAP↑ | NDS↑ | Road↑ | Vehicle↑ |
| Semantic Map | **11.29** | **16.28** | **26.26** | **66.48** | **29.76** |
| Vector Map | 12.07 | 15.73 | 25.84 | 65.17 | 28.38 |

Table 12: **Robustness to layout noise.** Performance under noisy layout shows graceful degradation across stages.

| Layout | OccGen | | ImgGen | 3DOD | | BEVSeg mIoU(%) | |
| --- | --- | --- | --- | --- | --- | --- | --- |
| | FID$^{3D}$ ↓ | F$^{3D}$ ↑ | FID↓ | mAP↑ | NDS↑ | Road↑ | Vehicle↑ |
| Clean | **258.8** | **0.778** | **11.29** | **16.28** | **26.26** | **66.48** | **29.76** |
| Noisy | 276.3 | 0.742 | 12.47 | 14.87 | 25.02 | 65.28 | 28.44 |

Table 13: **Inference efficiency** of each stage on a single RTX A6000.

| Stage | Steps | Time (s) | GPU (GB) |
| --- | --- | --- | --- |
| LayoutGen | 50 | 0.15 | 1.0 |
| OccGen | 20 | 3.25 | 7.7 |
| ImgGen | 20 | 2.30 | 7.0 |

# C  Additional Quantitative Results

## C.1  Effect of Spatial Conditioning

We evaluate the role of spatial conditioning using a *text-only* variant that removes all spatial inputs (layout maps, object boxes, and perspective maps) while retaining textual prompts. As shown in Table 10, the absence of spatial cues causes clear degradation in visual realism (FID ↑ 9.45) and spatial fidelity (Vehicle mIoU ↓ 22.27%), underscoring the importance of spatial conditioning for maintaining geometric coherence and consistent scene alignment.

## C.2  Effect of Layout Type

To examine different layout representations in our dual-mode controllability design, we compare two layout types: 1) *BEV semantic maps* for fine-grained spatial control and 2) *BEV vector maps* of object boxes and lanes for efficient customization. As shown in Table 11, both yield geometrically accurate and visually coherent scenes, while semantic maps provide stronger spatial priors with slightly better realism and downstream performance. This confirms that both layout types are fully compatible with our pipeline, enabling flexible and effective scene control.

## C.3  Robustness and Efficiency

To assess potential error accumulation in our cascaded generation pipeline, we conduct a noise-injection ablation by applying Gaussian perturbations (25% probability) to the initial layout, including 3D box centers and lane coordinates. As shown in Table 12, the pipeline degrades gracefully under noise, with only marginal drops in downstream metrics. This robustness arises from the multi-stage alignment design, where occupancy-rendered semantic and depth priors enforce geometric consistency, and overlap-aware extrapolation maintains spatial continuity.

We also report inference efficiency in Table 13. Each scene chunk is generated in about 6 seconds on a single RTX A6000 GPU, showing that our system achieves a strong balance between robustness and computational efficiency for large-scale scene synthesis.

## C.4  Effect of Retrieval-Augmented Generation

RAG enhances text-to-scene generation by expanding brief prompts into detailed scene descriptions through retrieving semantically related examples from a memory bank. This process transfers prior knowledge from similar scenes, improving layout accuracy and reducing user effort. Table 14 presents a human preference study in which ten participants evaluated 100 scene pairs generated with and without RAG across multiple criteria. The results show that RAG-based generation is consistently preferred (overall 77% vs. 23%), highlighting its

Table 14: **Human preference study** comparing scene generation w/ and w/o RAG.

| Criterion | RAG (%) | Non-RAG (%) |
| --- | --- | --- |
| Diversity | **87** | 13 |
| Realism | **82** | 18 |
| Controllability | **74** | 26 |
| Phys. Plaus. | **66** | 34 |
| **Overall** | **77** | 23 |

effectiveness in grounding prompts and producing more diverse, realistic, and controllable scenes.

# D   Additional Qualitative Results

## D.1   Conditional Occupancy and Image Generation

Figure 8 presents additional conditional generation results, where layout conditions are used to synthesize multi-view images and 3D occupancy. These results demonstrate the effectiveness of our approach in generating coherent multi-modal outputs conditioned on low-level layout inputs.

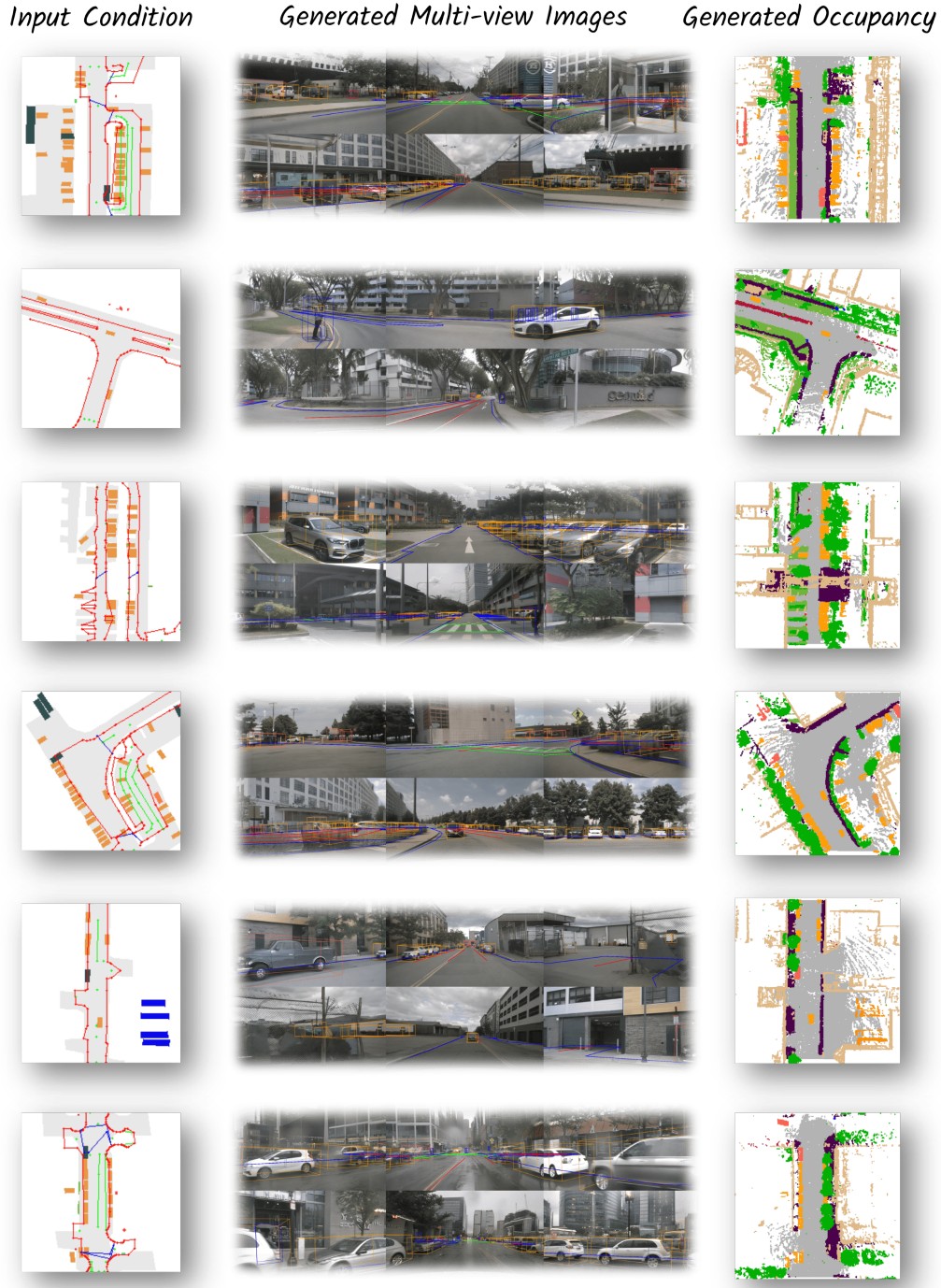

Figure 8: **Additional qualitative results of 𝒳-Scene on conditional occupancy and image generation.** These results demonstrate the model's ability to generate semantically consistent and structurally accurate multi-modal outputs conditioned on layout inputs across diverse urban scenes.

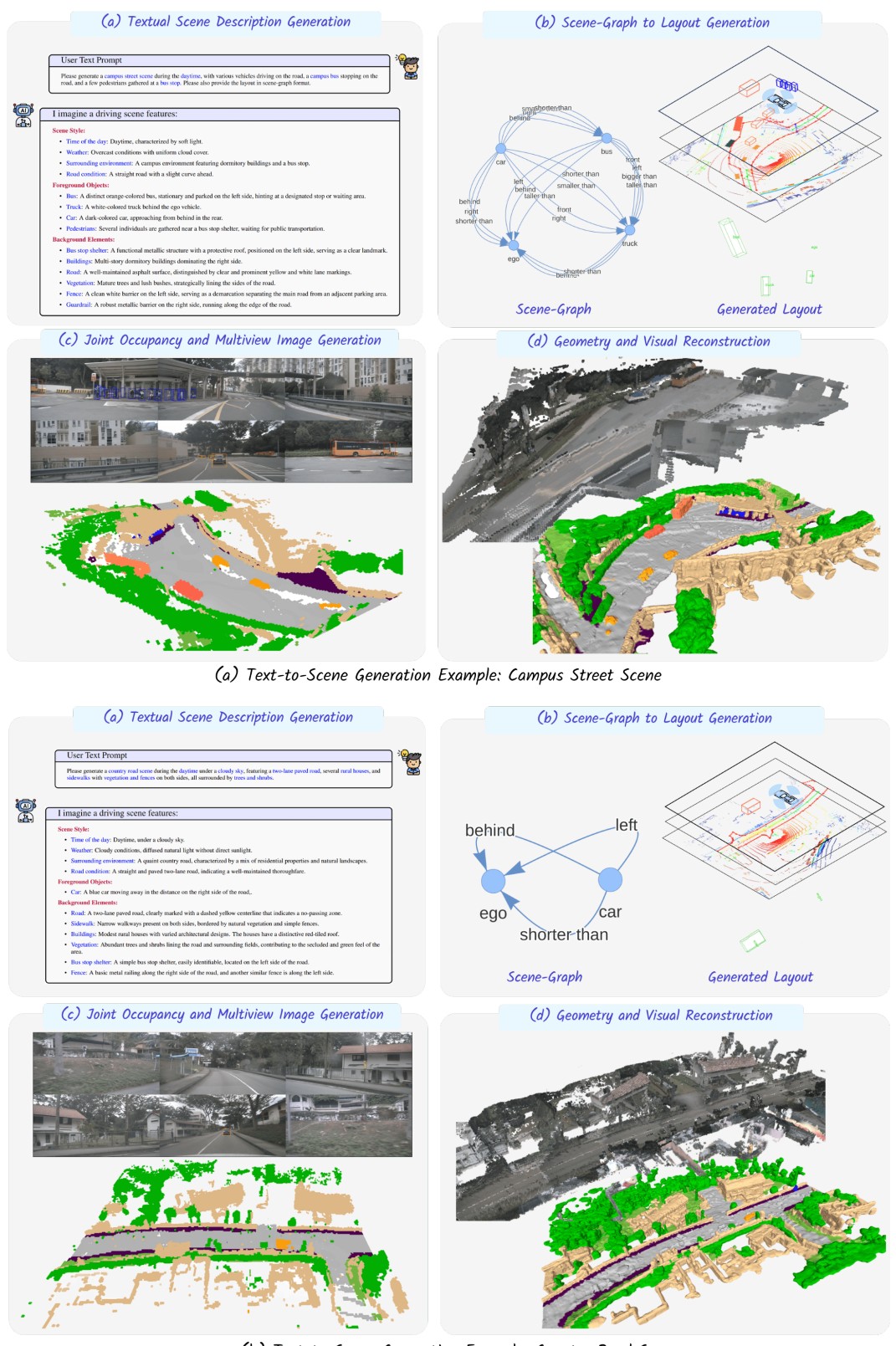

Figure 9: **Qualitative results of the text-to-scene generation pipeline of 𝒳-Scene**. Starting from a user prompt, the system generates a plausible scene description, constructs the corresponding layout, synthesizes consistent occupancy and multi-view images, and finally performs 3D reconstruction.

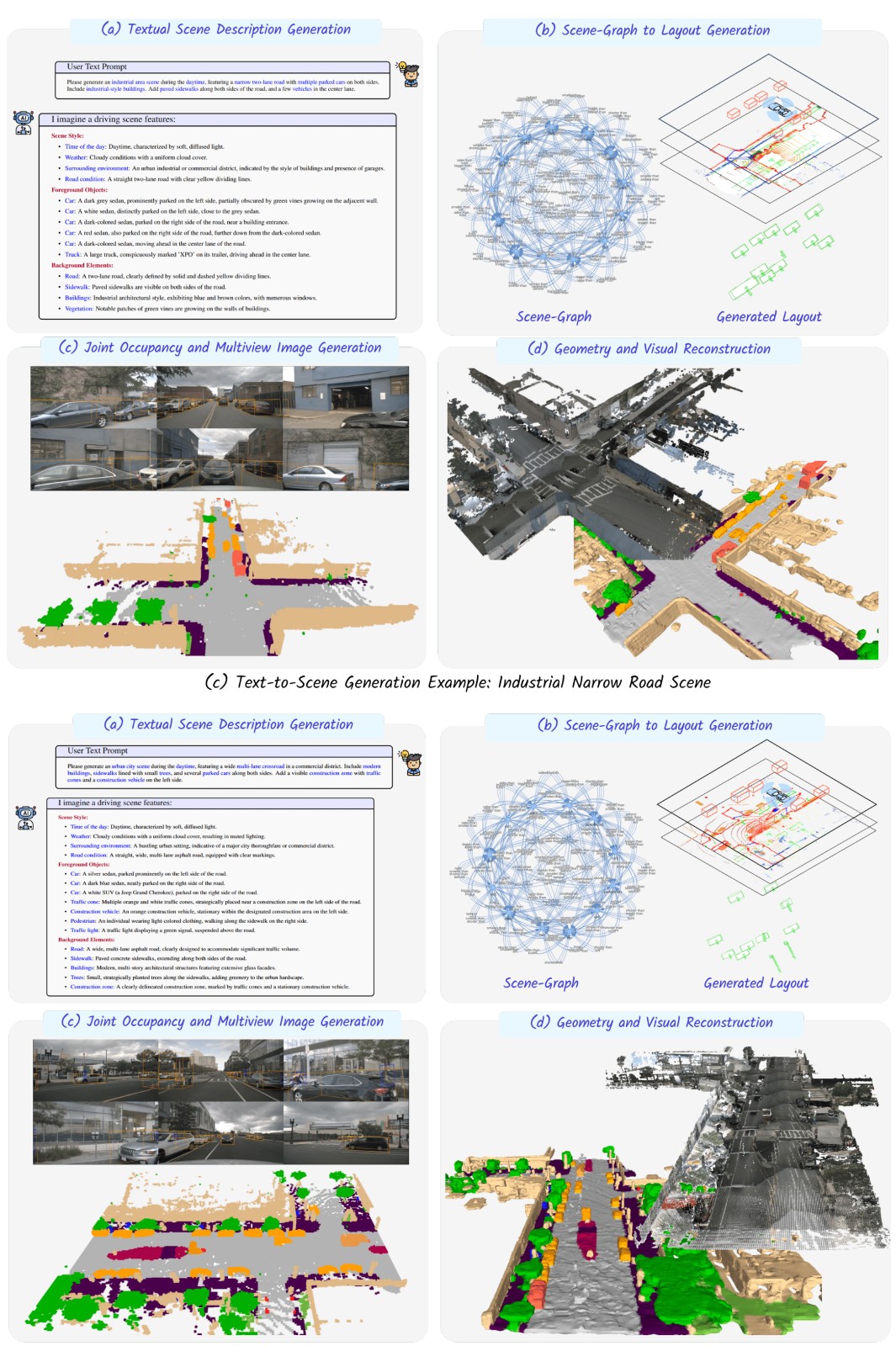

Figure 10: **Qualitative results of the text-to-scene generation pipeline of $\mathcal{X}$-Scene**. Starting from a user prompt, the system generates a plausible scene description, constructs the corresponding layout, synthesizes consistent occupancy and multi-view images, and finally performs 3D reconstruction.

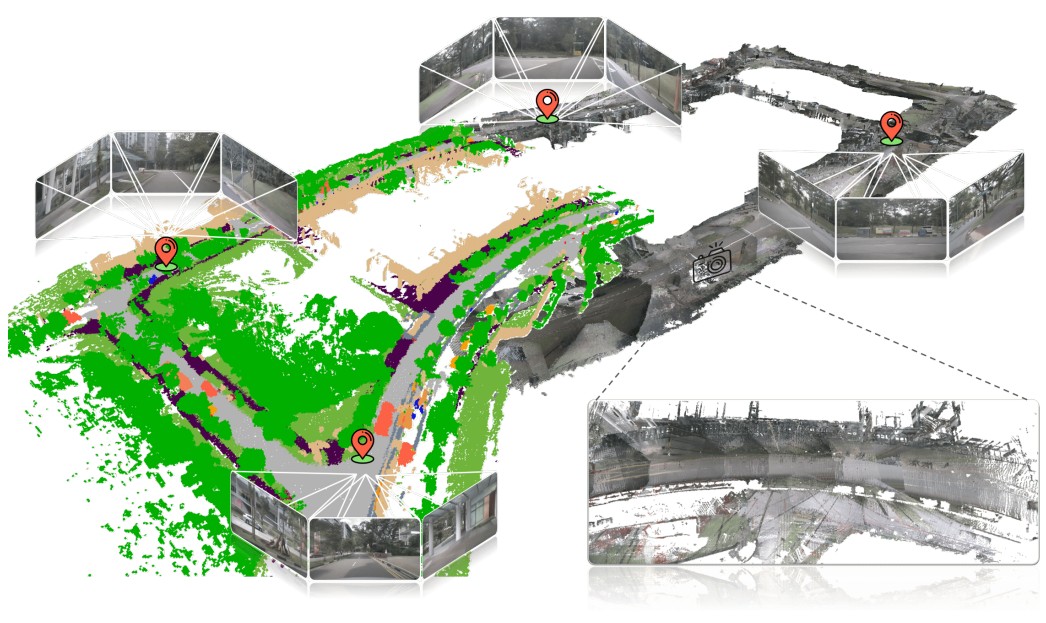

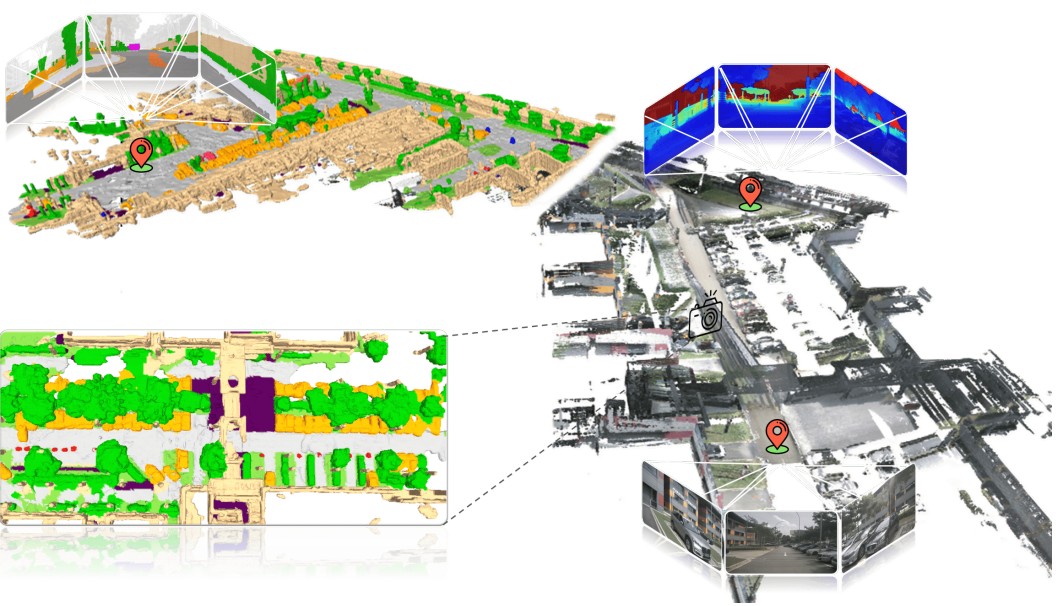

Figure 11: **Qualitative results of large-scale scene generation by $\mathcal{X}$-Scene.** The model extrapolates coherent occupancy fields and multi-view images across extended areas, enabling high-fidelity and complete 3D scene reconstruction. The generated scenes support novel view synthesis of RGB, depth, and occupancy, demonstrating both geometric consistency and high photorealistic quality at scale.

### D.2  Text-to-Scene Generation

Figure 9 and Figure 10 illustrate examples of the text-to-scene generation pipeline, which primarily consists of four steps:

- Textual scene description generation: Given a coarse user text prompt, the LLM leverages RAG to retrieve semantically relevant scene descriptions from the memory bank, then composes a plausible scene description encompassing scene style, foreground and background elements, and a textual scene-graph layout.
- Scene-graph to layout generation: The layout diffusion model uses the textual scene-graph to generate the corresponding layout, including object bounding boxes and lane lines.
- Joint occupancy and multi-view image generation: The occupancy and image diffusion models leverage the layout for geometry control and the text description for semantic control, generating a coherent and realistic 3D occupancy field and multi-view images.
- Geometry and visual reconstruction: Given the generated voxels and images, we reconstruct the 3D scene while preserving intricate geometry and realistic appearance, supporting various downstream applications.

These results demonstrate that the proposed text-to-scene pipeline is an effective and flexible method for driving scene generation.

### D.3  Large-Scale Scene Generation

Figure 11 illustrates the results of large-scale scene generation. The results show that our method can generate coherent, large-scale driving scenes through consistency-aware extrapolation. Moreover, the generated occupancy and images are fused and lifted for large-scale scene reconstruction, preserving both intricate geometry and realistic visual appearance. The reconstructed scenes support novel RGB, depth, and occupancy rendering.

## E  Potential Societal Impact & Limitations

In this section, we discuss the potential societal impact of our work and outline its possible limitations.

### E.1  Societal Impact

Our proposed framework, $\mathcal{X}$-Scene, for large-scale controllable driving scene generation holds significant potential for real-world societal impact. By unifying fine-grained geometric accuracy with photorealistic visual fidelity, $\mathcal{X}$-Scene enables the generation of highly realistic and semantically consistent 3D driving environments. This capability directly supports the development of safer and more efficient autonomous driving systems by enabling rigorous simulation and validation across richly diverse scenarios, including rare cases such as complex intersections, unexpected pedestrian behavior, and unusual road layouts. As a result, $\mathcal{X}$-Scene can accelerate the development cycle of autonomous vehicles, reduce reliance on costly and time-consuming real-world data collection, and improve safety standards, ultimately contributing to a reduction in traffic-related accidents and fatalities.

### E.2  Known Limitations

While $\mathcal{X}$-Scene offers a promising framework for large-scale controllable 3D scene generation, several limitations remain and warrant further investigation.

First, while $\mathcal{X}$-Scene supports dynamic 4D scene generation, the current autoregressive video diffusion framework is still limited in long-horizon synthesis. As the number of autoregressive iterations increases, errors in geometry and appearance may accumulate, leading to temporal drift and degraded motion consistency. Future work will focus on improving long-term temporal stability and mitigating error accumulation to achieve more robust and extended video generation.

Second, the scene description memory bank is currently built from the nuScenes dataset [93]. While this dataset provides a solid foundation, its limited geometric and semantic diversity may restrict the

range and realism of generated scenes. Incorporating additional datasets featuring a broader range of environments, weather conditions, and traffic patterns would enhance the system's generalization and scene richness.

Third, the occupancy generation pipeline depends on a fixed set of semantic categories predefined in the training data. As a result, introducing new object types or unseen classes requires retraining the model. This rigidity hinders adaptability in evolving or open-world settings. Future work could explore more extensible architectures that support incremental learning or open-vocabulary generation.

Addressing these limitations is essential for enhancing the realism, scalability, and applicability of $\mathcal{X}$-*Scene* in real-world simulation and data generation tasks.

# F  Public Resources Used

In this section, we acknowledge the public resources used, during the course of this work.

## F.1  Public Datasets Used

- nuScenes[1] ................................................... CC BY-NC-SA 4.0
- nuScenes-devkit[2] ........................................... Apache License 2.0
- Occ3D[3] ........................................................ MIT License

## F.2  Public Implementations Used

- MagicDrive[4] ................................................. Apache License 2.0
- SemCity[5] ........................................................ MIT License
- DynamicCity[6] ..................................................... Unknown
- DriveArena[7] ................................................. Apache License 2.0
- OccSora[8] .................................................... Apache License 2.0
- X-Drive[9] .................................................... Apache License 2.0
- MinkowskiEngine[10] .............................................. MIT License
- Torch-Fidelity[11] ............................................ Apache License 2.0
- Qwen2.5-VL[12] ............................................... Apache License 2.0
- UniScene[13] ................................................. Apache License 2.0

---

[1] https://www.nuscenes.org/nuscenes
[2] https://github.com/nutonomy/nuscenes-devkit
[3] https://tsinghua-mars-lab.github.io/Occ3D
[4] https://github.com/cure-lab/MagicDrive
[5] https://github.com/zoomin-lee/SemCity
[6] https://github.com/3DTopia/DynamicCity
[7] https://github.com/PJLab-ADG/DriveArena
[8] https://github.com/wzzheng/OccSora
[9] https://github.com/yichen928/X-Drive
[10] https://github.com/NVIDIA/MinkowskiEngine
[11] https://github.com/toshas/torch-fidelity
[12] https://github.com/QwenLM/Qwen2.5-VL
[13] https://github.com/Arlo0o/UniScene-Unified-Occupancy-centric-Driving-Scene-Generation

