# OpenReview forum: "X-Scene: Large-Scale Driving Scene Generation with High Fidelity and Flexible Controllability"
_NeurIPS.cc/2025/Conference — NeurIPS 2025 poster_

### Official Review · Reviewer_SavH · 2025-06-18

**Clarity:** 2
**Significance:** 3
**Originality:** 2
**Rating:** 4
**Confidence:** 5

**Summary:**

This paper proposes X-Scene, a novel pipeline for large-scale driving scene generation with high fidelity and flexible controllability. By dividing scene generation into 1) textual description enrichment, 2) textual scene-graph to layout generation, 3) joint occupancy and image generation, and 4) large-scale scene extrapolation and reconstruction, the proposed X-Scene can generate driving scenes with multiple modalities. Extensive experiments demonstrate the effectiveness.

**Questions:**

Check the Weakness part for details.

**Ethical Concerns:**

["NO or VERY MINOR ethics concerns only"]

**Final Justification:**

Overall, this is a solid paper with superior engineering efforts. Even if each component is relatively straightforward but inheriting existing practices, it is still fantastic to combine everything in a unified system. The authors further explain the unclear details missing in the initial draft, which makes it more beneficial for further progress. Thus, I decided to raise my initial score.

**Limitations:**

Check the Weakness part for details.

**Paper Formatting Concerns:**

Seems good.

**Quality:**

3

**Strengths And Weaknesses:**

- Strengths:
  - This is a solid work with superior engineering efforts, including LLM rephrasing, GNN for layout generation, occupancy generation, multi-view image generation, and occupancy & image outpainting.
  - The experiments thoroughly evaluate the performance of each component of X-Scene.
  - The authors provide the source code in the supplementary. I appreciate the authors for doing that.
- Weakness:
  - As a solid work with lots of effort, the explanation for each part is unfortunately quite unclear, including:
    - Line 132: How to describe spatial attributes via texts? Left or right?
    - Lines 141-144: The introduction of RAG is quite weird. Perhaps an ablation on using or not using RAG is necessary. Moreover, how to retrieve relevant samples for ICL given a specific input is unclear.
    - Lines 155-157: The definition of "layout" is quite unclear throughout the whole paper. According to Lines 155-157, the "layout" consists of object bounding boxes and road lines, while in Figure 1 and Table 3, it seems that the "layout" is more similar to BEV maps with semantic conditions.
    - Equation 2: I can see any conditions encoded in Sec. 3.1.
    - Equation 3: seems like an unconditional outpainting procedure, suggesting no supervision is applied for the outpainted regions.
    - Lines 216-230: Visual-Coherent Image Extrapolation seems to have no relationship with Geometry-Consistent Scene Outpainting? Otherwise, how to get the reference image at first?
    - Figure 4 caption: The camera poses supported by the image generative model are pre-defined by the nuScenes dataset, and thus, there is no way to generate novel views, even with occupancy conditions.
  - Novelty would be a controversial problem of this work, which mainly develops a pipeline for 3D street scene generation, while for each component, different existing works are utilized.
  - For a pipeline work with complicated component design, it would be essential to analyze the effects of error accumulation after each component.
- Overall, I think this is quite a solid work which combines recent developments of different areas of street scene generation with lots of effort. My concerns are mainly about the unclear paper writing, and I will consider increasing the score if the aforementioned questions are well-addressed after rebuttal.

---

> ### Author Rebuttal · Authors · 2025-07-31
>
> We thank the reviewer for recognizing the strength and scope of our work. This includes the **solid engineering effort**, **thorough experiments**, and the **inclusion of code** in the supplementary material.
>
> We address the remaining concerns that are primarily related to **clarity and technical explanation** through detailed responses and additional clarifications.
>
> > ### **Q1: Clarification on how spatial attributes are described via text in Line 132.**
>
> **A1:** Our textual spatial attributes cover three key dimensions:
>
> - **Relative Position**: e.g., *left, right, front, back, on top of* (e.g., *"the car is behind the truck")*;
> - **Relative Size**: e.g., *larger, smaller, taller, shorter* (e.g., *"the car is smaller than the truck")*;
> - **Heading Direction**: e.g., *heading left to right*, *facing front*, etc.
>
> These are parsed via predefined linguistic patterns into scene-graph layouts to encode both **inter-object spatial relationships** and **per-object size/orientation** (see supplementary Pages 7–8). We will clarify this in the final version.
>
> > ### **Q2: Clarification and justification of RAG usage and retrieval strategy for ICL.**
>
> **A2:** We clarify our motivation for using RAG and provide ablation results as follows:
>
> - **Motivation for using RAG:**
>    RAG is central to the *text-to-scene generation* mode. It transforms **brief, coarse** user prompts (e.g., *“a busy city intersection”*) into **detailed and structured descriptions** (see supplementary Pages 7–8, 10–11) by retrieving semantically relevant examples from the memory bank.
>    This mechanism not only **reduces user effort** but also improves generation quality by **transferring prior knowledge** from similar scenes.
> - **Ablation Study (with vs. without RAG):**
>    Since RAG is tailored for **user-prompted generation**, we conducted a **human preference study** to evaluate its effectiveness.
>    Using 100 prompt-matched scene pairs (w/ and w/o RAG), we asked 10 users to evaluate them across four criteria: **diversity, realism, controllability, and physical plausibility**.
>    As shown in **Table R1**, RAG-based outputs are consistently preferred. This confirms its effectiveness in transforming vague prompts into more diverse and realistic scenes.
> - **Retrieval mechanism for ICL:**
>    Our memory bank encodes both **high-level scene styles** (e.g., *sunny daytime city intersection*) and **low-level object attributes** (e.g., *construction area with barriers*). This enables flexible retrieval across **different levels of prompt granularity**.
>    At inference time, we encode the input prompt using `text-embedding-3-small` and perform **top-k retrieval based on cosine similarity**. The retrieved examples act as **few-shot demonstrations** to ground the LLM's generation in relevant prior scenes.
>
> **Table R1: Human Preference Study Comparing Scene Generation with vs. without RAG.**
> |Criterion|RAG Preferred (%)|Non-RAG Preferred (%)|
> |-|-|-|
> |Diversity|**87**|13|
> |Realism|**82**|18|
> |Controllability|**74**|26|
> |Physical Plausibility|**66**|34|
> |**Overall Preference**|**77**|23|
>
>
> > ### **Q3: Clarification on the definition of "layout": object boxes and road lines (Lines 155–157) vs. BEV semantic maps (Figure 1, Table 3)**
>
> **A3:** We clarify that our framework supports **two types of layouts** under a **dual-mode controllability design**:
>
> - **BEV vector maps** (object boxes + road lanes): These are generated from user-provided textual prompts via the scene-graph-to-layout diffusion module (as in Lines 155–157 and Fig. 2). This form supports the **text-to-scene generation mode** to offer a compact yet sufficient condition for efficient scene synthesis.
> - **BEV semantic maps**: These are directly specified by users for the **fine-grained spatial control mode** (as in Fig. 1 and Table 3), which enables detailed constraints over scene structure.
>
> Both layouts are compatible with our scene diffusion pipeline.
>
> - **Additional evaluation:** To further assess their impact on controllability and generation quality, we conduct an ablation study in **Table R2** using different layout types. The results show that both layout types support accurate scene generation, while semantic maps offer stronger spatial priors and lead to slightly better performance.
>
> We will revise the manuscript to better explain this **dual-mode layout design**.
>
> **Table R2: Ablation on layout types (BEV semantic map vs. vector map).**
> |Input Layout|FID↓|mAP↑|NDS↑|Rd. mIoU↑|Veh. mIoU↑|
> |-|-|-|-|-|-|
> |BEV semantic map|11.29|16.28|26.26|66.48|29.76|
> |BEV Vector map|12.07|15.73|25.84|65.17|28.38|
>
> > ### **Q4: Clarification on the absence of conditioning in Equation (2).**
>
> **A4:** We clarify the role of Equation (2) and where conditioning is applied:
>
> - **Purpose of Equation (2):** This equation defines the deformable attention in the triplane encoder. It is used purely for encoding input occupancy, and is not part of the conditional generation process.
> - **Where conditioning occurs:** Conditioning is applied later in the occupancy diffusion model $\epsilon^{occ}_{\theta}$ (**Lines 182–187**), which incorporates layout, box, and text conditions defined in Section 3.1.
>
> We will make this clearer in the final manuscript.
>
> > ### **Q5: Clarification on conditioning in Equation (3).**
>
> **A5:** Equation (3) defines an outpainting process guided by **implicit conditions**:
>
> - **Overlap-based conditioning:** The process uses the pre-generated **reference latent** $h_0^{ref}$ and an **overlap mask** $M$ to constrain the denoising of the new latent $h_t^{new}$, ensuring geometric continuity in the overlapped region (Fig. 3b).
> - **Implicit conditioning:** The overlapping region acts as an **implicit constraint in latent space** instead of explicitly copying pixels or voxels.
> - **No additional training:** This functions as an **inpainting-style procedure** within the diffusion model and requires no extra training.
>
> > ### **Q6: Clarification on the relationship between image extrapolation and occupancy outpainting.**
>
> **A6:** We clarify that image extrapolation is not an independent process but is **tightly coupled** with occupancy outpainting. Specifically, the generated occupancy is rendered into semantic and depth maps under camera views, which condition the image diffusion (see Fig. 1 and Lines 188-190).
>
> - **First chunk:** 3D occupancy is generated from noise via triplane diffusion and rendered into semantic/depth maps. These serve as conditions to guide the generation of multi-view images.
> - **Subsequent chunks**: Previously generated occupancy and images serve as references. The new occupancy is rendered again to guide the next image region.
>
> This design ensures consistent extrapolation of both **geometry and appearance** to enable coherent large-scale scene generation.
>
> > ### **Q7: Clarification on camera poses and novel view generation in Figure 4 caption.**
>
> **A7:** The reviewer is correct, and we appreciate the correction. In Figure 4, the rendered images are generated under the **pre-defined camera poses** from the nuScenes dataset. This setup ensures alignment with real-world views for consistent visualization and evaluation.
> We will revise the figure caption to clearly state this setup and avoid potential confusion.
>
> > ### **Q8: Clarification on the novelty of the proposed framework.**
>
> **A8:** We emphasize two key innovations that go beyond simple integration:
>
> - **Multi-Granular Control Mechanism**:
>   We introduce a **dual-mode control framework** that supports both (1) *text-prompted* scene generation for high-level usability and (2) *layout-guided* generation for precise control. This design balances usability and precision, enabling intuitive *high-level customization* (mode 1) and precise *low-level geometric control* (mode 2). To the best of our knowledge, **no prior work has explored such a flexible and unified control mechanism especially in the context of text-to-scene generation**.
> - **Unified Generation and Extrapolation Pipeline:**
>   Our framework jointly models *3D occupancy and multi-view image* generation, coupled with a *consistency-aware outpainting mechanism* for scalable scene extrapolation. This design ensures both geometric continuity and visual coherence. As a result, it enables realistic closed-loop and E2E simulation. This is an **improvement over existing methods that are limited to single-chunk or image-only generation**.
>
> These contributions collectively establish our X-Scene as a cohesive and powerful framework for large-scale and controllable 3D scene generation. They advance both the system’s capabilities and its practical applicability.
>
> > ### **Q9: Error accumulation in the cascaded pipeline.**
>
> **A9:** We address concerns about error accumulation and pipeline robustness through a **noise-injection ablation study**:
>
> - **Ablation setup:** We introduce Gaussian noise to the initial layout (i.e., 3D box centers and lane points) with 25% probability to simulate partial noise arising from imperfect layout predictions. We then assess the impact on occupancy generation, image synthesis, and downstream performance.
> - **Ablation results:** As shown in **Table R3**, the results **degrade gracefully** despite the injected noise. Particularly, in downstream detection (mAP ↓ 1.41) and BEV segmentation (Rd. mIoU ↓ 1.20, Veh. mIoU ↓ 1.32). This indicates the strong robustness of our pipeline.
> - **Why robust:** This resilience stems from our **multi-stage alignment design**: (1) occ rendered semantic/depth maps provide strong geometric priors for image generation; (2) overlap-aware extrapolation ensures spatial continuity across chunks.
>
> **Table R3: Ablation on Error Accumulation.**
> |Input Layout|OccGen FID³ᴰ↓|OccGen F³ᴰ↑|ImgGen FID↓|3DOD mAP↑|3DOD NDS↑|BEVSeg Rd. mIoU↑|BEVSeg Veh. mIoU↑|
> |-|-|-|-|-|-|-|-|
> |Clean Layout|**258.8**|**0.778**|**11.29**|**16.28**|**26.26**|**66.48**|**29.76**|
> |Noisy Layout|276.3|0.742|12.47|14.87|25.02|65.28|28.44|

---

> > ### Comment · Reviewer_SavH · 2025-08-04
> >
> > Thanks to the authors for the detailed rebuttal, which also backs up my initial review that this is indeed solid work with lots of engineering efforts, but the explanations are unfortunately unclear. I'm very glad to see lots of details are provided in the author's rebuttal, which I suggest the authors add to the final paper revision. I decided to increase my score to BA:)

---

> ### Author Response · Authors · 2025-08-04
>
> Dear Reviewer SavH,
>
> We sincerely thank you for your constructive comments and valuable suggestions, which greatly helped us present our method with more accurate and detailed explanations. Your feedback enabled us to clarify several important technical aspects, and we will incorporate these improvements into the revised version of the paper as recommended.
>
> We also truly appreciate your recognition of our work and efforts. Thank you again for the thoughtful and encouraging feedback :)

---

### Official Review · Reviewer_oXSL · 2025-06-29

**Clarity:** 3
**Significance:** 3
**Originality:** 4
**Rating:** 5
**Confidence:** 2

**Summary:**

This paper introduces X-Scene, an end-to-end pipeline for driving scene generation that supports various beneficial functions. The functions include converting text or user-drawn layouts into a scene graph and outpainting the scene chunk by chunk. The authors propose to use a triplane latent occupancy volume, multi-view dissusion model to generate the scene and pack the final output into a 3D Gaussian-splat scene for downstream usages. Extensive experiments demonstrate that X-Scene outperforms existing baselines in terms of generation quality, image size, and improvement in downstream perception tasks.

**Questions:**

Please see the above.

**Ethical Concerns:**

["NO or VERY MINOR ethics concerns only"]

**Final Justification:**

I appreciate the author's enthusiastic and impressive rebuttal. I have carefully read the fellow reviewers' comments and the response. All of my concerns have been well addressed. I will raise the score from *borderline accept* to *accept*, as I believe the research is well-designed, timely, and impactful.

**Limitations:**

The authors have adequately addressed the limitations and potential negative societal impact of their work.

**Paper Formatting Concerns:**

There are no major formatting issues in this paper.

**Quality:**

3

**Strengths And Weaknesses:**

Strenghts
---
- The paper is generally well-written and easy to follow.
- The research is timely and aims to improve the current generation in many aspects, including fidelity and controllability.
- Experiments are carefully designed and complete.

Weaknesses
---
Although I am not actively working in this field, I was unable to identify any critical weaknesses in the manuscript that would prevent me from giving a positive score. Below are some questions after reading through the paper.
- While I don't think it's the main point of the research, as the authors mentioned, the current method doesn't support video generation, which is critical for real usage such as closed-loop evaluation.
- Contributions may come from mostly an engineering unification rather than scientific novelty as key ideas including triplane, cross-modal diffusion, and latent extrapolation have already been widely used.
- It would be beneficial if the authors could provide a more in-depth discussion on the effectiveness of using RAG in their framework.
- How does outpainting behave after many hops? Does the artifact accumulate?
- Is there any reason for the performance discrepancy between Tables 5 and 7?

Additional Comments
---
- Missing references that the authors may want to add:
  - Wu et al. Text2lidar: Text-guided lidar point cloud generation via equirectangular transformer. In ECCV 2024.
  - Pan et al. Transfer Your Perspective: Controllable 3D Generation from Any Viewpoint in a Driving Scene. In CVPR 2025.

---

> ### Author Rebuttal · Authors · 2025-07-31
>
> The reviewer appreciated the **clear writing**, the **comprehensive and well-designed experiments**, and the **practical value** of our controllable, high-fidelity generation framework, which unifies scene graph conversion, triplane diffusion, multiview image generation, and chunk-wise outpainting, delivering **strong performance** in both image quality and downstream tasks.
>
> In response to the reviewer’s comments, we provide **additional results and clarifications** below to address video generation capability, architectural novelty, the effectiveness of RAG, outpainting robustness, and evaluation consistency.
>
> > ### **Q1: While it may not be the main focus of the paper, the lack of video generation limits the applicability such as for realistic closed-loop evaluation.**
>
> **A1:** While video generation was not the original focus, we have extended **X-Scene to support dynamic scene synthesis** by introducing an **autoregressive video generation framework** with temporal conditioning and attention, trained on 7-frame clips.
>
> - **FVD-based evaluation:** To assess temporal consistency, we adopt the **FVD metric** suggested by reviewers oXSL, w82Z, and sCXS.
> - **End-to-end evaluation:** We also use **UniAD** to test the effectiveness of the generated videos for E2E perception and planning tasks, including *3D object detection* (mAP, NDS), *BEV segmentation* (mIoU), and *trajectory planning* (L2 Distance, Collision Rate).
> - **Experiment results:** As shown in **Table R1**, X-Scene trained on 7-frame clips achieves **lower FVD** than the 16-frame-trained baseline *MagicDrive*. This indicates superior video fidelity with higher efficiency. Our X-Scene also **outperforms the baseline in end-to-end evaluation** via UniAD. This demonstrates its effectiveness for closed-loop simulation.
> - **Scalability:** Furthermore, our autoregressive design enables **long-horizon video generation** (e.g., over 100 frames) despite short-clip training. This showcases the strong scalability and sample efficiency of our design. Qualitative results will be included in the final paper (subject to rebuttal constraints).
>
> **Conclusion:** These results confirm that X-Scene now supports high-quality and temporally coherent **dynamic scenes generation**, making it well-suited for realistic end-to-end simulation. We will release the video generation code to support reproducibility.
>
> **Table R1: Evaluation of Video Generation Fidelity (FVD) and End-to-End Performance using UniAD.**
> |Method|VideoGen FVD↓|3DOD mAP↑|3DOD NDS↑|BEVSeg Lanes↑|BEVSeg Drivable↑|BEVSeg Divider↑|BEVSeg Crossing↑|L2\@1.0s↓|L2\@2.0s↓|L2\@3.0s↓|L2 Avg.↓|Col.\@1.0s↓|Col.\@2.0s↓|Col.\@3.0s↓|Col. Avg↓|
> |-|-|-|-|-|-|-|-|-|-|-|-|-|-|-|-|
> |MagicDrive[1]|217.9|12.92|28.36|21.95|51.46|17.10|5.25|0.57|1.14|1.95|1.22|0.10|0.25|0.70|0.35|
> |X-Scene (Ours)|**206.2**|**20.40**|**31.76**|**28.04**|**61.96**|**22.32**|**10.48**|**0.55**|**1.08**|**1.81**|**1.15**|**0.03**|**0.13**|**0.66**|**0.27**|
>
>
> > ### **Q2: Concern about the lack of scientific novelty beyond component integration.**
>
> **A2:** We emphasize two key innovations that go beyond simple integration:
>
> - **Multi-Granular Control Mechanism**:
>    We introduce a **dual-mode control framework** that supports both (1) *text-prompted* scene generation for high-level usability and (2) *layout-guided* generation for precise control. This design balances usability and precision, enabling intuitive *high-level customization* (mode 1) and precise *low-level geometric control* (mode 2). To the best of our knowledge, **no prior work has explored such a flexible and unified control mechanism especially in the context of text-to-scene generation**.
> - **Unified Generation and Extrapolation Pipeline:**
>    Our framework jointly models *3D occupancy and multi-view video* generation, coupled with a *consistency-aware outpainting mechanism* for scalable scene extrapolation. This design ensures both geometric continuity and temporal coherence. As a result, it enables realistic closed-loop and end-to-end simulation. This is an **improvement over existing methods that are limited to single-chunk or image-only generation**.
>
> These contributions collectively establish our X-Scene as a unified and effective framework for large-scale and controllable 3D scene generation. They advance both the capabilities of the system and its real-world applicability.
>
> > ### **Q3: Motivation and empirical evidence for using RAG in the framework.**
>
> **A3:** We clarify our motivation for using RAG and provide ablation results as follows:
>
> - **Motivation for using RAG:**
>    RAG plays a pivotal role in the *text-to-scene generation* mode. It transforms **brief, coarse** user prompts (e.g., *“a busy city intersection”*) into **detailed and structured** scene descriptions (see Pages 7–8 and 10–11 in the supplementary material). By retrieving semantically relevant examples from the memory bank, it enriches the input and enables accurate layout generation, guiding realistic scene synthesis.
>    This mechanism not only **reduces user effort** but also improves generation quality by **transferring prior knowledge** from similar scenes.
> - **Ablation Study (with vs. without RAG):**
>    As RAG is specifically designed for the **user-prompted generation** setting, we conducted a **human preference study** to evaluate its effectiveness.
>    We generate 100 scene pairs from identical user prompts, with and without RAG, and ask 10 users to compare them based on **diversity**, **realism**, **controllability**, and **physical plausibility**.
>    As shown in **Table R2**, RAG-enhanced outputs are consistently preferred. This confirms that RAG effectively transforms vague prompts into more **diverse** and **realistic** scenes through **effective retrieval and prior knowledge transfer**.
>
> **Table R2: Human Preference Study Comparing Scene Generation with vs. without RAG.**
> |Criterion|RAG Preferred (%)|Non-RAG Preferred (%)|
> |-|-|-|
> |Diversity|**87**|13|
> |Realism|**82**|18|
> |Controllability|**74**|26|
> |Physical Plausibility|**66**|34|
> |**Overall Preference**|**77**|23|
>
>
> > ### **Q4: Robustness of outpainting across multiple hops and potential artifact accumulation.**
>
> **A4:** Our design explicitly addresses error accumulation during outpainting through a **consistency-aware extrapolation mechanism** (Sec. 3.3, Fig. 3) that ensures stable generation across multiple hops:
>
> - **Local conditioning:** Each new chunk is conditioned on **overlapping regions** from the previous one, which serve as **spatial and appearance anchors** to maintain structural and visual continuity.
> - **Soft constraints:** We use **latent-level conditioning** instead of directly copying pixels or voxels from the previous chunk. This provides flexibility for correcting inconsistencies during denoising.
> - **Global regularization:** In addition, each hop also incorporates **global conditions** (layout, boxes, text prompts) to offer high-level semantic guidance across the scene.
> - **Diffusion robustness:** The denoising process in diffusion models further helps to **suppress minor artifacts** instead of propagating them.
>
> As shown in Fig. 4 and Page 12 of the supplementary material, our model consistently generates coherent large-scale scenes without noticeable degradation, even after many hops.
>
> > ### **Q5: Reason for the performance discrepancy between Table 5 and Table 7.**
>
> **A5:** The apparent performance difference stems from **different evaluation purposes**:
>
> - **Table 5 (Fine-tuning):** We assess the **training utility** of our generated data by using it to **fine-tune** downstream models. This demonstrates its effectiveness as a form of data augmentation.
> - **Table 7 (Direct testing):** We evaluate **inference-time compatibility** by **testing** how well pretrained models that are trained on real data perform on our generated data. This assesses the ability of real-data-trained models to generalize to generated inputs.
>
> Thus, Table 5 highlights how generated data benefits learning, and Table 7 reveals how it aligns with real data without any adaptation. We will clarify this distinction in the revised manuscript.
>
> > ### **Q6: Recommend adding related works (i.e., Text2LiDAR, Transfer Your Perspective)**
>
> **A6:** We appreciate the suggestion and agree that both works are highly relevant.
>
> - **Text2LiDAR** introduces a text-controllable framework for LiDAR point cloud generation. We extend this to text-conditioned *occupancy* and *multi-view video* generation for more comprehensive simulation.
> - **Transfer Your Perspective (TYP)** generates high-quality LiDAR from alternative viewpoints to simulate collaborative perception. Likewise, X-Scene focuses on large-scale scene generation with enhanced controllability and scalability.
>
> We will include appropriate citations and discuss these works in the revised manuscript.

---

> > ### Comment · Area_Chair_9EVd · 2025-08-06
> > **Read rebuttal, update final review and acknowledge**
> >
> > Dear reviewer, could you have read the rebuttal, update final review and share any remaining follow up questions if any? Also please acknowledge after you have done this. Thanks!

---

> > ### Comment · Reviewer_oXSL · 2025-08-06
> >
> > I appreciate the author's enthusiastic and impressive rebuttal. I have carefully read the fellow reviewers' comments and the response. All of my concerns have been well addressed. I will raise the score from *borderline accept* to *accept*, as I believe the research is well-designed, timely, and impactful.

---

> ### Author Response · Authors · 2025-08-07
>
> Dear Reviewer oXSL,
>
> Thank you very much for your thoughtful follow-up and positive evaluation. Your constructive feedback helped us clarify key aspects of our methodology and conclusions, and we will incorporate these clarifications into the revised version.
>
> We sincerely appreciate your recognition and support of our work.

---

### Official Review · Reviewer_sCXS · 2025-06-30

**Clarity:** 3
**Significance:** 1
**Originality:** 2
**Rating:** 4
**Confidence:** 3

**Summary:**

This paper introduces X-Scene, a framework for generating large-scale, high-fidelity, and controllable 3D driving scenes. X-Scene is designed as a multi-stage pipeline. It begins with a "Multi-Granular Controllability" module that allows user input via either high-level text prompts or low-level geometric layouts. This control signal guides the core generation engine, which first synthesizes a 3D semantic occupancy field using a triplane-based diffusion model, and then generates corresponding photorealistic multi-view images conditioned on the 3D geometry. To achieve large-scale generation, the system employs a "consistency-aware outpainting" mechanism to iteratively expand the scene chunk by chunk, ensuring both geometric and visual coherence. The final generated assets are lifted to 3D 3DGS. The authors demonstrate through extensive experiments that their method achieves best results in both generation quality and utility for downstream perception tasks.

**Questions:**

See weaknesses.

**Ethical Concerns:**

["NO or VERY MINOR ethics concerns only"]

**Final Justification:**

Thank you for the exceptionally thorough and compelling rebuttal. I am very impressed with the extensive work authors have done to address the initial concerns, which has strengthened the paper. Most of my concerns are very well addressed. I have raised my score to BA.

**Limitations:**

Yes

**Quality:**

2

**Strengths And Weaknesses:**

Strengths:

1: The hierarchical generation process (3D occupancy first, then 2D images) is a good way to ensure 3D consistency. The use of a triplane representation with a novel deformable attention mechanism is a nice touch that, as shown in the ablations, demonstrably improves reconstruction quality. The wealth of conditioning information used for the image diffusion model (rendered semantic/depth maps, perspective maps, object embeddings) to have better alignment between the generated geometry and appearance, a common failure point in other methods.

2: The experimental results are thorough. It also demonstrates significant gains when using its synthetic data to train downstream perception models. The ablation studies are excellent.

3: Paper is easy to follow.

Weaknesses:

1: The most significant weakness, which the authors also acknowledge, is that the framework generates static scenes.  Real-world driving is fundamentally dynamic. While generating high-quality static worlds is a critical first step, the lack of dynamics limits its immediate use for end-to-end planning and simulation.

2: X-Scene is a very complex, multi-stage cascading pipeline (LLM -> RAG -> Graph Diffusion -> Triplane Diffusion -> Image Diffusion). While powerful, this complexity raises concerns about reproducibility,  training/inference costs, and potential error propagation. How long is the inference time? Also a small error in the initial layout generation could lead to significant artifacts in the final rendered scene. The paper doesn't discuss how these errors are handled or how they might compound.

3: The paper claims large-scale generation via iterative outpainting. However, the practical limits of this approach are not discussed. What are the computational/memory constraints as the scene grows? Does the con

---

> ### Author Rebuttal · Authors · 2025-07-31
>
> We thank the reviewer for the detailed and constructive feedback. The reviewer appreciated **several key strengths of our work**, including the hierarchical 3D-to-2D generation strategy, deformable attention in triplane diffusion, and enhanced geometry-appearance alignment. The reviewer also acknowledged our **comprehensive experiments**, **strong downstream results**, and the **manuscript's clarity**.
>
> In response to the reviewer’s concerns, we provide **additional results and clarifications** below to address dynamics, error accumulation, computational cost, and outpainting scalability.
>
> > ### **Q1: While not the original focus, the lack of dynamic (video) scene generation acknowledged by the authors limits the framework's applicability to real-world end-to-end planning and simulation.**
>
> **A1:** While video generation was not the original focus, we have extended **X-Scene to support dynamic scene synthesis** by introducing an **autoregressive video generation framework** with temporal conditioning and attention, trained on 7-frame clips.
>
> - **FVD-based evaluation:** To assess temporal consistency, we adopt the **FVD metric** suggested by reviewers sCXS, w82Z, and oXSL.
> - **End-to-end evaluation:** We also use **UniAD** to test the effectiveness of the generated videos for E2E perception and planning tasks, including *3D object detection* (mAP, NDS), *BEV segmentation* (mIoU), and *trajectory planning* (L2 Distance, Collision Rate).
> - **Experiment results:** As shown in **Table R1**, X-Scene trained on 7-frame clips achieves **lower FVD** than the 16-frame-trained baseline *MagicDrive*. This indicates its superior video fidelity with higher efficiency. Our X-Scene also **outperforms the baseline in the end-to-end evaluation** via UniAD. This demonstrates its effectiveness for closed-loop simulation.
> - **Scalability:** Furthermore, our autoregressive design enables **long-horizon video generation** (e.g., over 100 frames) despite short-clip training. This showcases the strong scalability and sample efficiency of our design. Qualitative results will be included in the final paper (subject to rebuttal constraints).
>
> **Conclusion:** These results confirm that X-Scene now supports high-quality and temporally coherent **dynamic scenes generation**, making it well-suited for realistic end-to-end simulation. We will release the video generation code to support reproducibility.
>
> **Table R1: Evaluation of Video Generation Fidelity (FVD) and End-to-End Performance using UniAD.**
>
> |Method|VideoGen FVD↓|3DOD mAP↑|3DOD NDS↑|BEVSeg Lanes↑|BEVSeg Drivable↑|BEVSeg Divider↑|BEVSeg Crossing↑|L2\@1.0s↓|L2\@2.0s↓|L2\@3.0s↓|L2 Avg.↓|Col.\@1.0s↓|Col.\@2.0s↓|Col.\@3.0s↓|Col. Avg↓|
> |-|-|-|-|-|-|-|-|-|-|-|-|-|-|-|-|
> |MagicDrive[1]|217.9|12.92|28.36|21.95|51.46|17.10|5.25|0.57|1.14|1.95|1.22|0.10|0.25|0.70|0.35|
> |**X-Scene (Ours)**|**206.2**|**20.40**|**31.76**|**28.04**|**61.96**|**22.32**|**10.48**|**0.55**|**1.08**|**1.81**|**1.15**|**0.03**|**0.13**|**0.66**|**0.27**|
>
>
> > ### **Q2: Error accumulation in the cascaded pipeline, inference cost, and reproducibility.**
>
> **A2:** We address concerns about error accumulation and pipeline robustness through a **noise-injection ablation study**:
> - **Error accumulation ablation:**
>     1. **Ablation setup:** We introduce Gaussian noise to the initial layout (i.e., 3D box centers and lane points) with 25% probability to simulate partial noise arising from imperfect layout predictions. We then assess the impact on occupancy generation, image synthesis, and downstream performance.
>     2. **Ablation results:** As shown in **Table R2**, the results **degrade gracefully** despite the injected noise. Particularly, in downstream detection (mAP ↓ 1.41) and BEV segmentation (Rd. mIoU ↓ 1.20, Veh. mIoU ↓ 1.32). This indicates the strong robustness of our pipeline.
>     3. **Why robust:** This resilience stems from our **multi-stage alignment design**: (1) occ rendered semantic/depth maps provide strong geometric priors for image generation; (2) overlap-aware extrapolation ensures spatial continuity across chunks.
> - **Inference cost:** **Table R3** summarizes inference time and memory usage for each module. A single scene chunk takes \~6 seconds to generate with **moderate GPU consumption**.
> - **Reproducibility:** We provide detailed implementation info in Section A.2 of the supplementary material and will release code for full reproducibility.
>
> **Table R2: Ablation on Error Accumulation: Impact of Layout Noise on Generation Quality and Downstream Tasks.**
> |Input Layout|OccGen FID³ᴰ↓|OccGen F³ᴰ↑|ImgGen FID↓|3DOD mAP↑|3DOD NDS↑|BEVSeg Rd. mIoU↑|BEVSeg Veh. mIoU↑|
> |-|-|-|-|-|-|-|-|
> |Clean Layout|**258.8**|**0.778**|**11.29**|**16.28**|**26.26**|**66.48**|**29.76**|
> |Noisy Layout|276.3|0.742|12.47|14.87|25.02|65.28|28.44|
>
>
> **Table R3: Inference Costs for Each Module on a Single RTX A6000.**
> |Module|Denoise Steps|Time|GPU Memory|
> |-|-|-|-|
> |Layout Diffusion|50|150 ms|1.0 GB|
> |Occupancy Diffusion|20|3.25 s|7.7 GB|
> |Multiview Image Diffusion|20|2.30 s|7.0 GB|
>
>
> > ### **Q3: Memory and computation scalability of iterative outpainting.**
>
> **A3:** Our iterative outpainting design is **memory-efficient and scalable**, which enables large-scale generation without memory bottlenecks:
>
> - **Constant memory per chunk:** As detailed in Lines 202-215, each chunk is generated using only its **own latent state** (occupancy latent $\mathbf{h}^{\text{new}}$, image latent $\mathbf{x}^{\text{new}}$) and **overlapping context** from the previous chunk ($\mathbf{h}^{\text{ref}}$, $\mathbf{x}^{\text{ref}}$). No global scene context needs to be held in GPU memory.
> - **Efficient memory usage:** Once a chunk is generated, it is offloaded from GPU memory to RAM (as `NumPy arrays`). This allows GPU memory to be reused in subsequent iterations.
> - **Result:** This design ensures both memory usage and computation cost **per chunk to remain constant** (as Table R3) regardless of the overall scene size.
>
> We will clarify this efficiency in the final version.

---

> > ### Comment · Reviewer_sCXS · 2025-08-02
> >
> > Thank you for the exceptionally thorough and compelling rebuttal. I am very impressed with the extensive work authors have done to address the initial concerns, which has strengthened the paper. Most of my concerns are very well addressed. I have raised my score to BA.

---

> ### Author Response · Authors · 2025-08-02
>
> Dear Reviewer sCXS,
>
> Thank you very much for your thoughtful and constructive review. Your comments and suggestions have greatly helped us improve the quality and clarity of our work. We sincerely appreciate your patience, careful reading, and valuable feedback, which have been instrumental in strengthening the paper.
>
> We are also truly grateful for your positive evaluation and for raising your score, which further affirms our efforts. Thank you once again for your support and recognition.

---

### Official Review · Reviewer_w82Z · 2025-07-02

**Clarity:** 3
**Significance:** 3
**Originality:** 2
**Rating:** 5
**Confidence:** 4

**Summary:**

The authors describe a system based on multistage diffusion policies for synthesising driving scenarios from text prompts (and/or maps and boxes) to 360-degree images and semantic occupancy maps. The system can synthesise specially (but not temporarily) coherent scenarios and complete/expand/edit maps on a large scale.

**Questions:**

Here are some questions for the authors to comment on.
I have ordered them in priority order from my perspective.

Q1 - metric for spatial consistency in outpainting: while the completion and extension of the scenarios are discussed qualitatively, I think it would be interesting to evaluate the system quantitatively, especially at discontinuity points. One way would be to synthesise videos and use video metrics (FVD).

Q2 - baseline for pure text-based diffusion: how well would a pure text-based diffusion policy synthesise images of the scene from the given prompt? While most likely not spatially consistent or entirely following the map layout, it could provide hints on how the presented input affects the rendering qualities of such a model.

Q3 - low-level metric for image generation: FID gives important information about the semantic distribution of the generated scenes. Could the authors propose a low-level metric for evaluating the realism of the generated images, perhaps to the provided training set?

Q4 task metrics: please, can you explain why you limit your BEV segmentation evaluation to only road and vehicles?

**Ethical Concerns:**

["NO or VERY MINOR ethics concerns only"]

**Final Justification:**

The authors present a complete framework for driving scenario synthesis. The framework primarily utilises existing methodologies, yet it yields impressive evaluations and results.
The authors made an effort to address my questions and concerns.
The additional results and insights are valuable and support the claims of the work.
For this reason, I confirm my rating.

**Limitations:**

yes. I would also include any limitations in terms of the training data required to achieve such performances (volume, alignment, labels, etc.).

**Quality:**

3

**Strengths And Weaknesses:**

I will use the symbol (+) to represent strengths and (-) to represent weaknesses.

# Quality
(+) The work describes a substantial framework for scenario generation. The framework consists of multiple steps that enable high-level and low-level conditioning and editing, as well as faster rendering through 3DGS.

(+) The quality of the discussion, in terms of thoroughness in experiment design, is well-suited to demonstrating the system's performance.

# Clarity
(+) The work is well-written and structured.

(+) All components of the framework are clearly discussed and formulated.

(-) Some of the images contain quite small details that are difficult to appreciate on printed paper.

# Significance
(+) The work is interesting and covers a significant topic. The creation of scenarios for robotic tasks can lead to editable simulators that can enhance the policies trained on them, especially in edge cases not easily recorded in real datasets.

# Originality
(-) This work is crucially a system paper, which utilises multi-step diffusion policies to generate data. Except for minor adaptations, the concepts and elements are retrieved from previous works.

(+) This said, the scale and performance achieved are impressive.

(-) I suggest that the authors review `NeuralFloors: Conditional Street-Level Scene Generation From BEV Semantic Maps via Neural Fields` and `NeuralFloors++: Consistent Street-Level Scene Generation From BEV Semantic Maps`. Although it is unfortunately closed-source, it employs a similar triplane approach, combining NeRF and diffusion. From this, it would be interesting to take inspiration from metrics (FVD for consistency) and baselines (prompt-based diffusion policy to test how inputs affect the quality of reconstruction).

---

> ### Author Rebuttal · Authors · 2025-07-31
>
> We sincerely thank the reviewer for the constructive and encouraging feedback. The reviewer **appreciated our work's** substantial **framework design**, **comprehensive experimental setup**, **clear presentation**, and **strong practical significance**.
>
> The reviewer also raised valuable suggestions regarding evaluation metrics, ablations, and training requirements. As detailed below, we have **addressed all concerns** with targeted **experiments** and **clarifications.**
>
> > ### **Q1: Scene outpainting is discussed qualitatively; a quantitative evaluation of consistency using FVD (inspired by NeuralFloors and NeuralFloors++) is recommended.**
>
> **A1**: Thank you for the insightful suggestion.
>
> - **Incorporating prior work:** We have reviewed *NeuralFloors* and *NeuralFloors++* and agree they are highly relevant. We will cite them appropriately in the revised manuscript.
> - **FVD-based evaluation:** Following their protocol, we adopted **FVD**  to assess the temporal consistency of our scene outpainting. As shown in **Table R1**, our method achieves competitive performance, with strong **frame-to-frame coherence**.
>
> **Table R1: FVD Comparisons on Video Generation.**
> |Method|FVD↓|
> |-|-|
> |DriveGAN|502.3|
> |DriveDreamer|452.0|
> |WoVoGen|417.7|
> |MagicDrive|217.9|dd
> |DreamForge|209.9|
> |**X-Scene (Ours)**|**206.2**|
>
> > ### **Q2: Suggest evaluating a pure text-based diffusion baseline; although spatial inconsistency is expected, it can reveal how the input conditions affect generation quality.**
>
> **A2:** Thank you for the suggestion. We conducted an additional **text-only generation** experiment to assess the role of spatial conditioning:
>
> - **Experiment setup:** We excluded all spatial conditions (i.e., layout maps, boxes, and perspective maps) while retaining only the textual prompt.
> - **Experiment results**: As shown in **Table R2**, compared to our full model, the text-only variant shows a substantial decline in spatial coherence (mAP ↓ 14.15, Veh. mIoU ↓ 22.27). However, it still maintains a plausible semantic distribution (FID ↑ 9.45).
> - **Interpretation:** Note that downstream evaluations are performed using downstream models trained on GT-aligned real images. The generated outputs exhibit spatial mismatches in the absence of spatial constraints, which naturally lead to degraded performance.
> - **Conclusion:** These results demonstrate that while text prompts convey high-level semantics, **spatial conditions are crucial** for ensuring both *visual realism* and *spatial fidelity*, especially for downstream utilization.
>
> **Table R2: Ablation on Text-only Generation.**
> |Variants|ImgGen FID↓|3DOD mAP↑|3DOD NDS↑|BEVSeg Rd. mIoU↑|BEVSeg Veh. mIoU↑|
> |-|-|-|-|-|-|
> |Full Model|**11.29**|**16.28**|**26.26**|**66.48**|**29.76**|
> |Text Only|20.74|2.13|5.34|28.32|7.49|
>
>
> > ### **Q3: Suggest incorporating low-level metrics to assess image realism beyond FID’s semantic-level evaluation.**
>
> **A3:**  Thank you for the suggestion. We have already included **low-level realism metrics** in the main paper (Tables 1–3) and further expanded the evaluation in **Table R3**:
>
> - **Existing evaluations (Tables 1–3 of the main paper):**
>     1. **Table 1** evaluates images generated from *GT bounding boxes and HD maps* using pretrained downstream models for **3D object detection** (mAP, NDS) and **BEV segmentation** (mIoU). High scores indicate that both *foreground objects* and *background layouts* are visually realistic and well-aligned with real data.
>     2. **Tables 2 and 3** assess the impact of generated images when used as **data augmentation** to fine-tune downstream models. Performance gains over the baseline demonstrate the **low-level fidelity and practical utility** of our generated data.
> - **Additional validation (Table R3):**
> We further conduct an **end-to-end evaluation** using UniAD, covering *3D object detection* (mAP, NDS), *BEV segmentation* (mIoU), and *trajectory planning* (L2 Distance, Collision Rate). These results provide further evidence through **low-level metrics**, to show that the generated images are both *realistic and practical* for downstream applications.
>
> **Table R3: Evaluation of Video Generation Fidelity (FVD) and Additional Low-level End-to-end Metrics using UniAD.**
> |Method|VideoGen FVD↓|3DOD mAP↑|3DOD NDS↑|BEVSeg Lanes↑|BEVSeg Drivable↑|BEVSeg Divider↑|BEVSeg Crossing↑|L2\@1.0s↓|L2\@2.0s↓|L2\@3.0s↓|L2 Avg.↓|Col.\@1.0s↓|Col.\@2.0s↓|Col.\@3.0s↓|Col. Avg↓|
> |-|-|-|-|-|-|-|-|-|-|-|-|-|-|-|-|
> |MagicDrive[1]|217.9|12.92|28.36|21.95|51.46|17.10|5.25|0.57|1.14|1.95|1.22|0.10|0.25|0.70|0.35|
> |X-Scene(Ours)|**206.2**|**20.40**|**31.76**|**28.04**|**61.96**|**22.32**|**10.48**|**0.55**|**1.08**|**1.81**|**1.15**|**0.03**|**0.13**|**0.66**|**0.27**|
>
> > ### **Q4: Justification for evaluating BEV segmentation solely on road and vehicle classes.**
>
> **A4:** Thank you for the question. We **follow established practice** in the main evaluation, and provide **extended results** in Table R3:
>
> - **Standard protocol:** We follow standard practice by using **CVT** [83] as the downstream model for BEV segmentation. CVT is pre-trained specifically to predict **road** and **vehicle** categories in the BEV view. As a result, prior works including ours typically report metrics on these two classes to ensure comparability.
> - **Extended evaluation:** We additionally use the end-to-end **UniAD** model for BEV segmentation (see **Table R3**) to provide a more comprehensive assessment. Unlike CVT, UniAD supports a **broader set of semantic classes** including **Lanes, Drivable Area, Dividers, and Pedestrian Crossings**. This extended evaluation complements the CVT-based results and further validates the semantic realism and spatial alignment of our generated scenes under **richer label settings**.
>
> > ### **Q5: Clarify training data requirements (e.g., volume, alignment, labels).**
>
> **A5:** Thank you for the question. Below we clarify the dataset and preprocessing used for training:
>
> - **Dataset:** We use nuScenes dataset with 1,000 diverse driving scenes under varying weather, lighting, and traffic conditions. Each 20-second scene includes approximately 40 keyframes. This results in around 40,000 annotated samples, where each sample provides 360° images from six cameras, 3D occupancy, bounding boxes, and HD maps. Following the standard protocol, we use 700 scenes for training and 150 for validation.
> - **Video upsampling:** For video generation, we apply ASAP interpolation to upsample to 12 Hz, producing ~240 frames per scene since annotations are available only at 2 Hz. This denser supervision supports more consistent training for temporally coherent video synthesis.

---

> ### Comment · Reviewer_w82Z · 2025-08-05
>
> Thank you for the detailed response. I believe the additional results and insights will strengthen the final version.

---

> ### Author Response · Authors · 2025-08-05
>
> Dear Reviewer w82Z,
>
> We sincerely thank you for the thoughtful feedback and recognition. Your constructive suggestions have directly helped us extend our experimental results and deepen our analysis. We will incorporate the additional findings and insights into the final version of the paper.
>
> Once again, we truly appreciate your positive evaluation and valuable input.

---

### Author Response · Authors · 2025-08-05
**General Response**

**Dear Reviewers, ACs, and SACs,**

We sincerely thank you for your time, insightful feedback, and constructive suggestions!

---

We are encouraged by your **recognition** of the scope, technical rigor, and practical value of our work:

- Reviewer `w82Z` praised our *“substantial framework”* with *“high-level and low-level conditioning”*, and commended our *“thorough experiment design”* and *“clear structure”*.
- Reviewer `sCXS` highlighted our *“hierarchical generation process”*, *“novel triplane attention”*, *“excellent ablation studies”*, noting our method as *“significantly effective for downstream perception”*.
- Reviewer `oXSL` appreciated our *“well-written paper”*, *“carefully designed experiments”*, and *“practical pipeline design”*.
- Reviewer `SavH` recognized our *“solid engineering efforts”*, *“multi-modal generation pipeline”*, and *“complete experimental evaluation”*.

---

In response to your helpful suggestions, we have made the following **clarifications and improvements**:

- **Video Generation Capability**
(*Suggested by Reviewers `w82Z`, `sCXS`, `oXSL`, and `SavH`*)

    - We extended the pipeline to support **dynamic scene synthesis** via an **autoregressive video generation framework**.
    - Evaluated with **FVD** and UniAD metrics, our method outperforms strong baselines such as *MagicDrive*.
    - The design enables *long-horizon video generation* (100+ frames) from 7-frame training, achieving strong temporal consistency and scalability.

- **Quantitative Evaluation & Robustness**
(*Suggested by Reviewers `w82Z`, `sCXS`, and `SavH`*)

    - We adopted FVD to quantify scene outpainting quality.
    - Additional ablations show:
        - *Text-only generation*: leads to major performance drops, highlighting the need for spatial conditioning.
        - *Injecting noise into layout inputs*: results in only mild degradation, indicating pipeline robustness.

- **RAG Effectiveness & Retrieval Mechanism**
(*Suggested by Reviewers `w82Z`, `oXSL`, and `SavH`*)

    - We clarified the *role of RAG* in expanding *brief prompts* into *structured scene descriptions*.
    - A *human preference study* (100 pairs, 10 users) shows RAG consistently improves *realism*, *diversity*, and *controllability*.
    - Retrieval is based on text embedding and cosine similarity from a structured memory bank.

- **Layout Design and Conditioning**
(*Suggested by Reviewer `SavH`*)

    - Clarified the **dual-mode layout design**:
        - *Vector layouts*: generated via scene-graph-to-layout diffusion (text-to-layout).
        - *Semantic BEV maps*: directly specified for precise control.
    - Ablation confirms both are effective, with semantic maps providing slightly better spatial alignment.

- **Beyond Integration: Key Innovations**
(*Suggested by Reviewers `sCXS`, `oXSL`, and `SavH`*)

    - A *multi-granular control mechanism* that supports both *textual* and *layout-based* generation.
    - A *unified occupancy and image outpainting pipeline* that ensures geometric and visual consistency.
    - These components together enable *scalable*, *controllable*, and *realistic* simulation environments.

- **Writing & Presentation Clarifications**
(*Suggested by Reviewer `SavH`*)

    - Clarified how *spatial attributes* (position, size, direction) are parsed and encoded.
    - Explained:
        - Equation (2) encodes occupancy features, not part of conditional generation.
        - Equation (3) defines latent-level inpainting using overlap-aware soft constraints.
    - Further clarified that image generation is directly coupled with occupancy outpainting.

---

We would like to re-emphasize the **key contributions** of this work:

- **X-Scene**, a unified and scalable framework for *controllable driving scene generation*, now extended to *dynamic video generation*.
- A **dual-mode control system** supporting both text prompts and layout-based constraints.
- A **multi-stage pipeline** for joint 3D occupancy and multi-view image generation, with consistency-aware outpainting.
- **Extensive experiments** on *fidelity*, *temporal consistency*, and *downstream usability*, validated via UniAD and human preference studies.


---

As the **Author–Reviewer Discussion phase** continues through August 8 (Anywhere on Earth), we warmly welcome any further questions or suggestions and remain fully available for continued discussion.

Thank you again for your thoughtful reviews and kind support.

*Warmest regards,*
The Authors of Submission 2794

---

### Decision · Program_Chairs · 2025-09-17

**Decision:**

Accept (poster)

**Comment:**

This work presents a novel scene generation framework that is both highly realistic and controllable. The framework combines multiple stages of generation and supports both low-level and high-level control. Post-rebuttal, all reviewers reached consensus in favor of acceptance, and the AC concurred.

The AC recommends acceptance, with the expectation that all promised changes and additional experimental results will be incorporated into the final version.